# Isotope-encoded spatial biology identifies plaque-age-dependent maturation and synaptic loss in an Alzheimer's disease mouse model

Jack I. Wood [1,2,13], Maciej Dulewicz [1,3,13], Alicja Szadziewska [1], Sophia Weiner[1,4,5], Junyue Ge [1], Katie Stringer[1,2], Sneha Desai[1,2], Lydia Fenson[1], Diana Piotrowska[1], Gunnar Brinkmalm [1,6], Srinivas Koutarapu[1], Haady B. Hajar [2], Kaj Blennow [1,6,7,8], Henrik Zetterberg [1,4,6,9,10,11], Damian M. Cummings[2], Jeffrey N. Savas [12], Frances A. Edwards[2] & Jörg Hanrieder [1,3,4,5] ✉

Understanding how amyloid beta (Aβ) plaques develop and lead to neurotoxicity in Alzheimer's disease remains a major challenge, particularly given the temporal delay and weak correlation between plaque deposition and cognitive decline. This study investigates how the evolving pathology of plaques affects the surrounding tissue, using a knock-in Aβ mouse model (*App^NL-F/NL-F*). We combined mass spectrometry imaging with stable isotope labeling to timestamp Aβ plaques from the moment of their initial deposition, enabling us to track their aging spatially. By integrating spatial transcriptomics, we linked changes in gene expression to the age of the plaques, independent of the mice's chronological age or disease stage. Here we show that older plaques were associated with reduced expression of synaptic genes. Additionally, when correlated with structure-specific dyes, we show that plaque age positively correlated with structural maturation. These more compact and older plaques were linked to greater synapse loss and increased toxicity.

The mechanisms underlying Alzheimer's disease (AD) pathogenesis are still not fully understood. AD pathology is characterized by extracellular Aβ plaques and intracellular hyperphosphorylated tau tangles with dementia[1]. The prevailing model of AD pathogenesis is that changes in Aβ metabolism precipitate a damaging cascade upstream of tau pathology and eventual neurodegeneration[2]. The relevance of Aβ peptides in AD has seen a recent resurgence as Aβ targeting antibodies lecanemab and donanemab have seen positive outcomes in Phase 3 trials as well as FDA approval[3,4]. However, Aβ levels are elevated in AD brains years before synapse loss, circuit dysfunction, neurodegeneration, and impaired cognition[1]. It is

therefore important to capture the disease at the earliest events of Aβ plaque formation. An understanding of Aβ pathogenesis is complicated by the heterogeneous presentation of plaque pathology including cored plaques but also diffuse—as well as vascular plaques[5,6]. At a cellular level, Aβ has been found to alter synaptic vesicle dynamics[7], where exo—and endocytosis machinery are already altered in the very early stages of Aβ pathology onset[8] and the probability of transmitter release has been shown to be altered even before plaques can be detected[9,10]. Clinically, this is in line with the hypothesis that Aβ pathology precipitates several decades before cognitive symptoms occur. Most importantly, these early effects of

Aβ aggregation, including synaptic changes, precede the development of mature Aβ plaques that can be detected by PET imaging or by CSF analysis[11–13]. Efforts to monitor Aβ plaque development directly have relied on in vivo cranial window imaging. This method has shown that plaque growth occurs gradually, taking place over weeks to months[14–16]. However, the use of dyes in these studies limits the ability to track plaques during their early stages, as the dyes only attach to plaques that already exhibit a fibrillar structure. Additionally, cranial window imaging can monitor only a limited number of cortical plaques at a time. Therefore, exactly how plaques develop over time from initial deposition, the extent of their diversity and their relation to toxicity or homeostatic response of surrounding neuronal circuits remains unclear[17].

Novel developments within chemical imaging, such as mass spectrometry-based imaging (MSI)[18–20] together with stable isotope labeling greatly increase our ability to study these events. Imaging of stable isotope labeling kinetics (iSILK) allows the delineation of plaque formation dynamics in a spatiotemporal manner while maintaining high molecular specificity[21].

Herein, we make use of these developments and interface the iSILK approach with both hyperspectral microscopy and spatial transcriptomics. Importantly, the dynamic nature of this approach allows us to track precipitating plaque pathology in the $App^{NL-F/NL-F}$ ($App^{NL-F}$) AD mouse model, which mirrors the age-associated and gradual plaque development observed in human Alzheimer's disease. We demonstrate that not only do plaques cause local synaptic damage as has been previously[22] demonstrated but individual plaques cause increasing synaptic damage as they persist and mature in the brain. Together, the spatial biology approach presented here provides a detailed picture of Aβ-aggregation, plaque formation, and maturation in concert with the associated cellular and molecular response in and around individual plaques.

## Results

### iSILK delineates spatial and structural patterns of plaque formation and maturation

Initial amyloid aggregation and subsequent plaque formation are dynamic processes, and difficult to capture whilst tracking across chronological ageing. We previously demonstrated the potential of stable isotope labeling and MSI to assess relative timing of Aβ plaque formation within individual $App^{NL-G-F}$ mice[21]. While these initial experiments were key for establishing the technique and its feasibility, there are limitations with respect to the $App^{NL-G-F}$ model whereby the Arctic mutation of $App$ alters the chemical properties of Aβ leading to an increased aggregation propensity not seen in sporadic AD. This results in a rapid and early plaque onset with a pronounced formation of cored deposits[23]. Therefore, by building on the previous findings from $App^{NL-G-F}$ mice, we here aimed to follow Aβ aggregation in another model, $App^{NL-F}$ mice, with more slowly progressing plaque pathology[23]. Although rare plaques have been reported to appear as early as 6 months[23], we reliably detect the first plaques at 9 months. Thereafter, plaque load and size along with microglial response increase steadily to 18 months and reach a plateau by 24 months[10,23]. This gradual increase in plaque pathology with age more closely resembles human disease. The use of the $App^{NL-F}$ knock-in mouse in this study also offered a biochemical advantage: since this model predominantly produces non-modified Aβ1-42, as evidenced by LC-MS/MS proteomic analyses of immunoprecipitated Aβ peptides (Supplementary Fig. 1A-F), as well as purified amyloid fibrils from $App^{NL-F}$ mouse brain extracts, it enables specific iSILK-based monitoring of Aβ1-42 using matrix assisted laser desorption ionization- time of flight mass spectrometry (MALDI-ToF MS).

For the iSILK experiments, $App^{NL-F}$ mice ($n = 7$) were labeled with $^{15}$N spirulina diet from age 6–10 months (PULSE period, 4 months, Fig. 1A) and either culled immediately at 10 months old ($n = 3$, Fig. 1A,

Experimental Design 1), or after an 8-month CHASE (washout) at 18 months old ($n = 4$, Fig. 1A, Experimental Design 2). Here, $App^{NL-F}$ mice at 10 and 18 months of age represent early and established stages of plaque pathology, respectively (Fig. 1B).

For Experimental Design 1, continuous labeling was used to identify whether the labeling design was implemented in time to ensure that secreted Aβ peptides were labelled prior to extracellular deposition and precipitation in plaques[23]. MALDI MSI was then used to determine stable isotope enrichment in plaque-associated, deposited Aβ peptides in situ. This approach enables us to image plaques by spatially measuring Aβ1-42 species (Fig. 1C) and correlating the isotope enrichment of the deposited Aβ peptides with the different plaque structures such as the center and periphery, thereby investigating the temporal development of these features during plaque formation.

At 10 months of age, the sparse deposition of Aβ plaques in the cortex contained solely $^{15}$N-enriched Aβ1-42. Background peptide signals are largely absent due to the use of formic acid during sample preparation, which enhances the signal-to-noise (S/N) ratio for selective amyloid imaging (Supplementary Fig. 2A-D). Importantly, it was observed that all Aβ1-42 peptides detected showed $^{15}$N incorporation (Supplementary Fig. 3A-C). This ensured that the onset of the labeling at 6 months old was sufficient to incorporate the $^{15}$N label into APP prior to plaque pathology onset, observed at 9 months.

After establishing the feasibility of the iSILK approach in $App^{NL-F}$ mice, we aimed to address various aspects of plaque maturation and heterogeneity using a multimodal imaging approach. First, we investigated intra-plaque heterogeneity (center vs. periphery) across different brain regions using MALDI MSI to better understand the chronological deposition of Aβ1-42 within plaques. Second, we assessed structural plaque maturity over time through correlative hyperspectral imaging and MALDI MSI (Fig. 1D). Third, we explored how plaque maturation influences the local, plaque-associated transcriptome by combining MALDI MSI with spatial transcriptomics (Fig. 1E). Finally, selected gene expression changes were validated by targeted immunohistochemistry (IHC) near plaques (Fig. 1F).

To address the question of intra-plaque heterogeneity (center vs. periphery), the relative $^{15}$N content of these plaque structures was evaluated. For this, we isolated plaque region of interest (ROI) spectral data of hippocampal and cortical deposits. We compared the isotopologue signature of plaque-associated Aβ1-42 that encodes the degree of labeling, as indicated by the shift in mass as well as peak broadening (Fig. 1A, G Experimental Design 1 and 2). Here, to estimate the degree of $^{15}$N incorporation and hence the difference in label content, we compared the full width at half maximum (FWHM), a measure of peak breadth, of the curve fitted to the Aβ1-42 isotopologue pattern, attributed to the comparably low mass resolution in linear mode (LP, Fig. 1E). As illustrated in Fig. 3B, peak broadening, and thereby an increase in the FWHM, corresponds to elevated $^{15}$N content and therefore plaque age. The tradeoff with linear mode was needed to ensure sensitive peptide detection at high spatial resolution.

Based on the labeling design, this alteration in the isotopologue signature indicates earlier or later deposition. Here, plaques in 10-month-old $App^{NL-F}$ mice (PULSE age 6–10 months, no CHASE) displayed lower $^{15}$N content for Aβ1-42 in the center compared with the periphery (Fig. 1H, $P < 0.001$). This suggests that previously deposited Aβ1-42 species have less $^{15}$N at the center, while later secreted Aβ1-42 deposits at the periphery. In 18-month-old animals (Experimental Design 2, PULSE age 6–10 months, 8-month CHASE), the pattern was reversed due to the significant deposition of unlabeled Aβ1-42 during the CHASE period (Fig. 1I, K, $P < 0.001$).

Both experiments suggest indeed that in precipitating amyloid plaque pathology in $App^{NL-F}$ mice the center of the plaque represents the first deposited Aβ. Over time, this initial structure of the plaque gets gradually compacted into a fibrilized core upon homogenous plaque-wide deposition of less aggregated Aβ during the plaque growth phase.

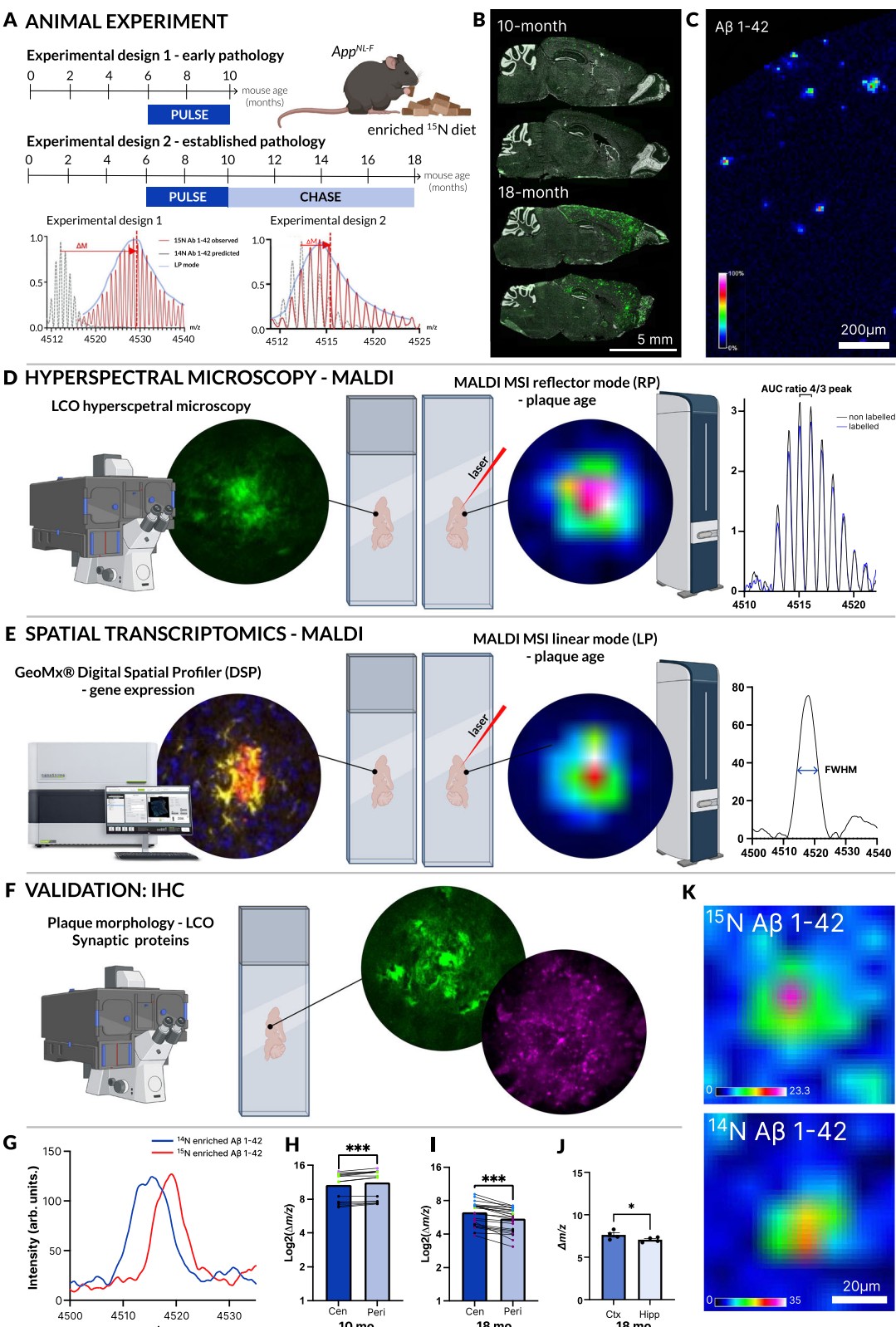

While these first analyses identified deposition patterns within plaques, iSILK further allows us to delineate plaque formation sequences across different brain regions. This is important as it allows for a description of how plaque pathology spreads throughout the brain and in what sequence it affects vulnerable brain regions. Similar to the plaque heterogeneity analyses, isotope incorporation was determined from the isotopologue patterns of Aβ1-42 in plaque features detected in the cortex and hippocampus as described above. The results show that the degree of isotope content, as expressed as the centroid of the fitted isotopologue distribution curve, was higher in cortical plaques compared to hippocampal plaques (Fig. 1J, P < 0.05). This indicates that the cortex is the primary location of initial plaque formation, which is in line with immunohistochemical studies described for this model as well as previous iSILK data obtained for *App*$^{NL-G-F}$ mice[21].

**Fig. 1 | iSILK delineates spatial and structural patterns of plaque formation and maturation. A** PULSE-CHASE iSILK paradigm. Experimental Design 1: PULSE period ($^{15}$N diet) between 6-10 months of age. Experimental Design 2: PULSE period between 6-10 months of age, CHASE period ($^{14}$N diet) between 10–18 months of age. Resulting Aβ1-42 MALDI MS isotopologue pattern that is right-shifted (Δm) due to increasing $^{15}$N incorporation. **B** Representative images of plaque load from GeoMx whole slide scans, repeated on four independent whole-brain slices at 18-months and three at 10-months. **C** MALDI MSI single ion image of Aβ1-42 in cortex section. **D** Schematic overview of the correlative hyperspectral imaging and MALDI MSI experiment. $^{15}$N enrichment (nitrogen index) was calculated as the AUC ratio of the 4th to 3rd peak in the Aβ1-42 isotopologue pattern. Higher values indicate greater 15 N incorporation. **E** Schematic overview of the correlative spatial transcriptomics (GeoMx) and MALDI MSI experiment. Stable $^{15}$N enrichments (nitrogen index) corresponding to plaque age was calculated by extracting the FWHM of the Aβ1-42 peak, where a broader peak indicates increased $^{15}$N incorporation and higher age. **F** Schematic overview of the validation experiment. Plaque morphology was evaluated by LCO hyperspectral imaging. IHC of selected proteins were correlated with plaque age, as evaluated by hyperspectral imaging. **G** Representative spectra from MALDI MSI showing the $^{14}$N and $^{15}$N-enriched Aβ1-42 m/z peak. **H** Aβ1-42 mass analysis comparing the plaque center (Cen) vs. the periphery (Peri) in 10-month-old mice ($p = 0.00017$), (**I**) in 18-month-old mice ($p = 0.00000077$), and (**J**) differences between cortex and hippocampus ($p = 0.022$). **H, I** Linear Mixed Model accounting for across animals and repeated measures, point color indicates animal, 15 replicates over $n = 3$ m mice and 22 replicates over n = 4 m mice, respectively. **J** Two-sided Paired t-test, 22 replicates over $n = 4$ m mice, data presented as mean ± SEM. **K** Representative MALDI MSI image of $^{15}$N and $^{14}$N enriched Aβ1-42 distribution in plaques in 18-month-old mice. Parts of the figure created in BioRender. Szadziewska, A. (https://BioRender.com/4qpojxz) Image in (**E**) provided by Bruker Spatial Biology. Significance levels: *** $P < 0.001$, **$P < 0.01$; *$P < 0.05$. Source data are provided as a Source Data file. FWHM full width at half maximum, RP reflector mode, LP linear mode.

## Amyloid plaque maturation is characterized by continuous fibrilization with age

Since we identified that the center of the deposits observed at both 10 and 18 months represents the oldest part of the plaque, we also aimed to investigate the structural maturation of plaques as they age in addition to changes in gene expression. Therefore, we integrated the iSILK MSI experiments with functional fluorescent microscopy (Fig. 1D) using structure-specific amyloid probes: luminescent conjugated oligothiophenes (LCOs)[24], allowing for the identification of morphologically heterogeneous amyloid structures within the Aβ plaques (Fig. 2A). This is enabled by the difference in affinity of the two LCO probes, q- FTAA and h-FTAA, towards amyloid aggregates. Specifically, q-FTAA preferentially binds to mature and compact beta-pleated aggregates, while h-FTAA binds to less compact, yet still beta-pleated aggregates[25]. Due to their different emission profiles[26], the LCO probes can be spatially delineated using hyperspectral fluorescent microscopy. Here, the ratio of the LCO maxima (500 nm for q-FTAA / 580 nm for h-FTAA) is used to express preferential binding of either of the two LCO probes used, whereby an increase in 500 nm intensity is indicative of increased q-FTAA binding and therefore, increased structural maturity of the amyloid fibrils[27,28] (Fig. 2B). As only few plaques were detected by MALDI in the 10-month-old animals (Fig. 1A, Experimental Design 1), of which all were cortical, we focused on 18-month-old $App^{NL-F}$ animals (Fig. 1A, Experimental Design 2) and compared plaques in the cortex to those in the hippocampus. Similar to the iSILK data showing that cortical plaques are older than those in the hippocampus, our results demonstrate that cortical plaques also exhibit higher 500/580 nm ratios at their centers (Fig. 2C) in both 9-month (Fig. 2D) and 18-month-old (Fig. 2E) mice, suggesting that they are more aggregated. This data thus pointed to a correlation of the hyperspectral ratio with plaque age, in which higher q/h ratios are associated with increasing plaque maturity. To further test this hypothesis, we performed a single plaque correlation analysis of correlative iSILK/LCO data collected from plaques detected across two sequential tissue sections (Fig. 1D, Supplementary table 1-2). The results indeed showed a positive correlation (R = 0.64, $P < 0.01$, Fig. 2F) of label incorporation and LCO emission ratio. This correlation could be replicated within individual mice (R = 0.8–0.9 $P < 0.05$; Supplementary Fig. 4A–C), indicating a biologically robust association between both variables. Together, these data show a direct correlation of the time course of individual plaques with biophysical measures of amyloid aggregation within individual animals. These results confirm that in these mice, plaques appear to continuously mature in terms of amyloid fibrillization at the plaque core.

## iSILK-guided spatial transcriptomics shows changes in synaptic-, metabolic-, and immune-associated gene expression with plaque age

Utilizing the iSILK technique to track plaque age, we conducted single plaque spatial transcriptomics to monitor gene expression changes associated with plaque age, independent of the mouse's chronological age. The spatial transcriptomics platform GeoMx® Digital Spatial Profiler was selected over other sequencing techniques due to its ability to target plaque-specific gene expression changes, offering a more spatially resolved technique for AD pathology-associated alterations compared to the more commonly used RNA sequencing methods[29]. The areas of illumination (AOIs) chosen on the spatial transcriptomics platform were selected by identifying double positive (MALDI and IHC) plaques across two serial sections, with the MALDI MSI images imported and overlaid onto the IHC image for precise mapping (Fig. 3A, and Supplementary Table 3-4). In accordance with our labeling design, older plaques displayed an increase in $^{15}$N-enriched Aβ1-42 signal ($m/z_{centroid} = 4523$) compared to younger plaques, as well as a broader FWHM (Fig. 3B) Aβ1-42 peak, corresponding to an elevated nitrogen index as evaluated by MALDI in LP mode (Fig. 1E).

To identify associations between the plaque age (MALDI iSILK) and whole transcriptome-wide gene expression, we performed correlation analyses across all plaques between the MALDI MS peak FWHM i.e., nitrogen index, a proxy for plaque age (Fig. 1E), and quantile normalized counts for all genes. Correlation analyses were conducted separately across plaques from 10-month-old (Supplementary Fig. 5, Supplementary Table 3) and 18-month-old mice (Supplementary Fig. 6, Supplementary Table 4), respectively. Detailed information on relevant MALDI MSI spectral as well as spatial transcriptomics data stratified by each individual mouse can be found in Supplementary Table 5. Volcano plots for both 10-month (Fig. 3C) and 18-month-old (Fig. 3D) mice demonstrated that gene expression levels had significant positive and negative correlations with increasing plaque age.

In 10-month-old animals (Fig. 1A, Experimental Design 1), functional annotation of correlated gene expression revealed that plaques maturity was associated with upregulation of genes involved in T cell activation, differentiation, and adaptive immune responses (e.g., *H2-K1, Lgals3, Anxa1, Irf1, Il6, Ly9, Cd55b, Socs5, Nfkbid*; Fig. 3E). Although gene ontology enrichment was dominated by adaptive immunity, several dysregulated genes, including *Anxa1* (Fig. 3F) and *Axl* (Fig. 3G), are well-established markers of microglial activation. In contrast, synaptic gene expression was negatively associated with plaque age (Fig. 3H) and included, among others, *Nptx2* (Fig. 3I), *Nptxr, Grin2a, Camk2a, Nrgn, Dlgap1, Dlgap3*, and *Dlg4* (Fig. 3J).

In 18-month-old animals (Fig. 1A, Experimental Design 2), the gene ontology of correlated gene expression shows that older plaques are linked to an increase in transcriptional activity of genes involved in metabolic processes (Fig. 3K), such as *Gdnf, Ggct, Optn, Gabarapl1, Csf1* (Fig. 3L), *Ctsd* (Fig. 3M), *Pi4k2a*, and channel activity (*Kcnrg, Ano10, Chrnb4, Kcna4, Kcnab1*) compared to younger plaques in the same animal. Additionally, older plaques showed enrichment for genes related to chromatin organization and

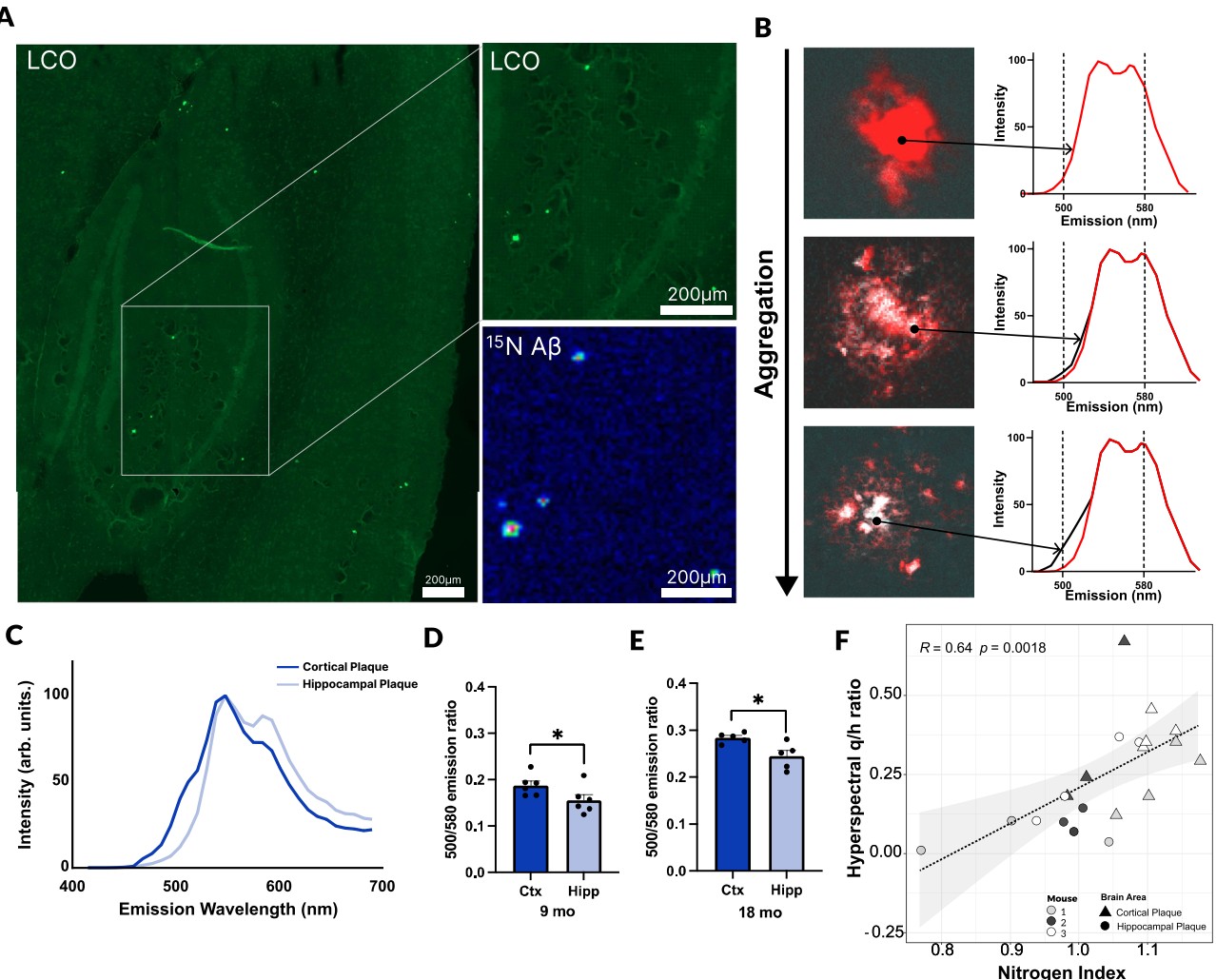

**Fig. 2 | Amyloid plaque maturation is characterized by continuous fibrilization with age. A** Representative images showing plaque detection via luminescent conjugated oligothiophenes (LCOs) staining and MALDI-MSI, repeated on 3 (18-months) independent MSI/LCO whole-brain slice pairs. LCOs bind to amyloid fibrils and reflect their structural conformation through spectral shifts. **B** Schematic illustrating increasing 500 nm intensity resulting from q-FTAA binding as plaques aggregate. **C** Averaged LCO emission spectra comparing hippocampal and cortical plaques. **D, E** Analysis of the emission ratio between the 500 nm q-FTAA peak and

the 580 nm h-FTAA peak, in cortex vs. hippocampus for (**D**) 9-month-old, $n = 6$ ($p = 0.041$) and (**E**) 18-month-old mice, $n = 5$ ($p = 0.0317$). Two-sided Mann-Whitney test. **F** Two-sided Pearsons's correlation analysis of the 500/580 nm hyperspectral ratio with the nitrogen index (Aβ1-42 isotopic peak ratio, RP mode) for $n = 3$ m 18-month-old mice. Individual mice are color-coded with the shape indicating either cortical (triangle) or hippocampal (dot) plaques. Data presented as mean ± SEM. Significance levels: **$P < 0.01$; *$P < 0.05$. Source data are provided as a Source Data file.

epigenetic regulation (*Dyrk1a, Setd1a, Trim28*), cytoskeletal remodeling and GTPase activity (*Fgd1, Arhgef5, Coro1c*), immune cell recruitment and adhesion (*Itga4, Ccl21a, Fut9*), and protein trafficking and secretion (*Xpo1, Eif6, Lamp2*) (for a detailed overview of all GO terms, refer to the Shiny App in the Data Availability section). In contrast, increasing age of plaques correlated with a concomitant decrease in gene activity predominantly associated with post synapses and synaptic membrane (Fig. 3N), including *Shisa6, Chrna9, Drd1, Nptx2* (Fig. 3O), *Gria1, Grin2b, Calb1, Dlgap1, Dlgap3, Dlg4* (Fig. 3P). Additionally, gene expression related to both disease-associated microglia (DAM) and disease-associated astrocytes (DAA) correlated positively with plaque age in 10-month-old (*e.g., Anxa1, Lgals3, H2-K1, Ly9*; Supplementary Fig. 7A) as well as 18-month-old (C*sf1, Ctsd, Fabp3*; Supplementary Fig. 7B) mice. This resonates with the central role of both cell types in plaque maturation and suggests not only a spatial but also temporal component to microglial and astrocytic reactivity to plaques. To identify gene-gene correlation clusters across plaque maturation stages in both 10- and 18-month-old mice, we performed a weighted gene co-

expression network analysis (WGCNA, Supplementary Fig. 8). In line with our results reported above, gene clusters most strongly associated with plaque age were related to either synaptic (negative correlation with plaque age) or inflammatory processes (positive correlation with plaque age).

To ensure that the observed regulatory changes are not, in fact, driven by systematic differences in AOI (area of illumination) size, the number of cells surrounding each plaque, or changes in cell type heterogeneity, we correlated plaque age with corresponding AOI size, cell count (Supplementary Fig. 9) and GFAP immuno-stained area, a marker of astrogliosis (Supplementary Fig. 10). No significant correlation was observed for either metric ($P > 0.05$), suggesting that differences in gene expression are unlikely to arise from varying cell density or composition surrounding aging plaques.

Using synaptic hub genes[30] and disease-associated genes[31], we compared expression profiles between previously generated spatial transcriptomics data and RNA-seq results from bulk hippocampal tissue analysis[32] (Supplementary Fig. 11). While spatial transcriptomics effectively captured expression changes associated

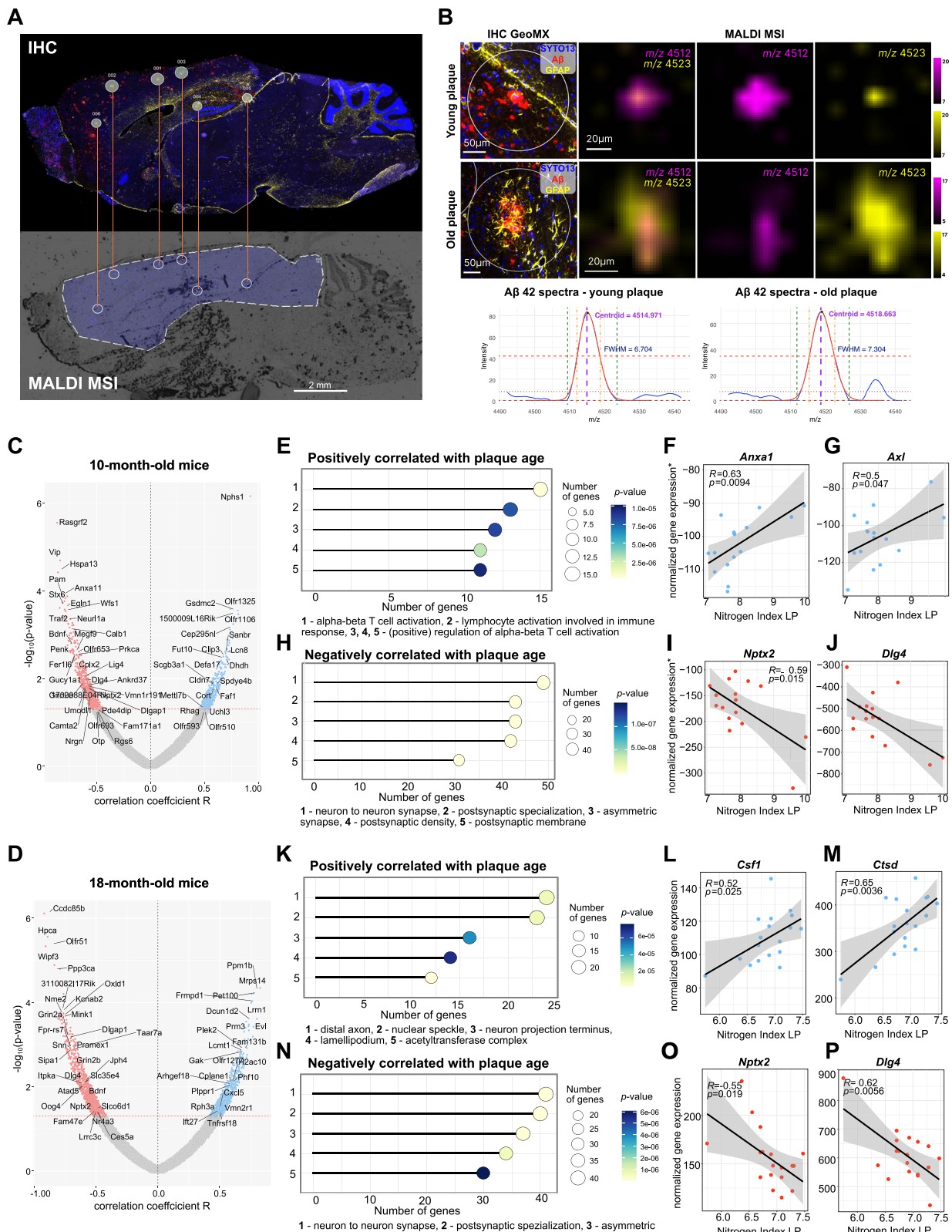

with Alzheimer's pathology, bulk RNAseq was less sensitive in detecting these differences. Additionally, incorporating plaque age, as enabled by iSILK, reveals further changes related to plaque maturity that are overlooked when only considering plaques at a given chronological age[32]. This underscores the significant advantages of spatial transcriptomics in addressing diseases with strong spatial characteristics, such as AD.

## Differences in synapse loss and toxicity revealed by LCO-defined plaque types

As amyloid aggregation/LCO probe binding is a factor of plaque age, we adapted our hyperspectral imaging approach to categorize amyloid plaques depending on the presence of q-FTAA and h-FTAA, representing different stages of plaque maturation. For this, we used linear unmixing of the hyperspectral LCO data utilizing reference

**Fig. 3 | iSILK guided spatial transcriptomics shows changes in synaptic-, metabolic-, and immune-associated gene expression with plaque age.**
**A** Matched Aβ plaques from GeoMx IHC and MALDI MSI. **B** IHC signal of two GeoMx AOIs, representing a young and old plaque, with corresponding intact peptide MS spectra and MS single ion images showing the spatial distribution for non-labelled/ monoisotopic (m/z = 4512) and $^{15}N$-enriched (m/z = 4523) Aβ1-42. An increase in $^{15}N$-enriched Aβ1-42 signal and broader Full Width at Half Maximum (FWHM) of the Aβ1-42 peak indicate advanced plaque age. **C, D** Volcano plots showing gene expression correlations with increasing plaque age in (**C**) 10-month-old $App^{NL-F}$ mice (Experimental Design 1) and (**D**) 18-month-old $App^{NL-F}$ mice (Experimental Design 2). Genes significantly correlated (non-FDR-adjusted $P < 0.05$) with plaque age are depicted in blue (positive correlation) and red (negative correlation). Correlation strength is indicated by the Pearson correlation coefficient $R$ (x-axis). **E** Gene ontology analyses of positively correlated genes with plaque age in 10-month-old mice. **F, G** Example genes, *Anxa1* (**F**) and *Axl* (**G**), positively correlating with plaque age in 10-month-old mice. **H** Gene ontology analyses of negatively correlated genes with plaque age in 10-month-old mice. **I, J** Example genes, *Nptx2* (**I**) and *Dlg4* (**J**), negatively correlating with plaque age in 10-month-old mice. **K** Gene ontology analyses of positively correlated genes with plaque age in 18-month-old mice. **L, M** Example genes, *Csf1* (**L**) and *Ctsd* (**M**), positively correlating with plaque age in 18-month-old mice. **N** Gene ontology analyses of negatively correlated genes with plaque age in 18-month-old mice. **O, P** Example genes, *Nptx2* (**O**) and *Dlg4* (**P**), negatively correlating with plaque age in 10-month-old mice. *$y$ axis has been inverted to account for the difference in labeling design between 10-month- (Experimental Design 1) and 18-month-old mice (Experimental Design 2) to enable a direct comparison of the directionality of gene expression changes with plaque age. **E, H, K, & N** GO enrichment by Fisher's exact test with Benjamini-Hochberg correction FDR < 0.05. **F, G, I, J, L, M, O, P** Two-sided Pearsons correlation. For all 10-month data: 16 replicates (AOI) from $n = 3$ m mice; for all 18-month data: 18 replicates (AOI) from $n = 4$ mice.

spectra of pure q-FTAA and h-FTAA to separate the probes into distinct channels (Fig. 4A). Additionally, an Aβ42 antibody served as a general Aβ marker. We focused exclusively on 18-month-old animals, as the 10-month-old group presented too few hippocampal plaques to yield a sufficient dataset. We identified three populations of structurally distinct plaque types depending on the positivity of the LCO probes: Aβ + h + q + , Aβ + h + q-, and Aβ + h-q-, here arranged in descending order of aggregation maturity (Fig. 4B). In our paired LCO/iSILK results, we established that continuous plaque aging is associated with increased q+ staining i.e., amyloid fibrilization (Fig. 2F). This suggests that Aβ + q + h + , Aβ + q + h-, and Aβ + q-h- represent distinct stages of plaque maturation. Therefore, this static imaging approach allows us to study plaque ages dissected from chronological age within the same animal using fluorescent microscopy-based correlated LCO imaging and immunohistochemistry within the same section.

We explored the frequency of different hyperspectral imaging-defined plaque types in a larger number of non-metabolically labelled $App^{NL-F}$ mice, culled at 9 months ($n = 7$), 14 months ($n = 5$) or 18 months ($n = 6$) of age. On average, Aβ + h + q+ plaques were the largest in area accounting for 41% of Aβ-positive regions in the hippocampus, followed by Aβ + h + q- plaques, which occupied 24%. Of note, many small ( <100mm²), LCO-negative (Aβ + h-q-) plaques were detected in the hippocampus of 18-month-old animals, which constituted 35% of the plaque-positive area (Supplementary Fig. 12). Those small immature plaques showed low Aβ intensity and are difficult to capture in MALDI MSI, particularly spanning across two consecutive 12 μm sections for correlative MALDI MSI/GeoMx or MALDI/LCO experiments, limiting these analyses to LCO pos. plaques. Moreover, when examining these plaque types over the lifespan of the mice, we observed that Aβ + h + q- and Aβ + h + q+ plaques together make up 90% of the Aβ-positive area in the hippocampus of younger, 9-month-old $App^{NL-F}$ mice (Supplementary Fig. 12). By 14 months, the proportion of Aβ-positive area occupied by Aβ + h-q- plaques rose to 26% (53% of plaques by number), increasing further to 36% (78% of plaques by number) by 18 months (Supplementary Fig. 12). This indicates that as the mice age, the number of plaques seeded without a fibrillar core increases. Together these findings support the notion that in this mouse model at pathology onset, plaques form as small cores, which continuously mature and fibrilize, respectively, through progressing amyloid deposition.

Considering the observation above that increasing plaque age is associated with changes in transcription of synaptic genes (Fig. 3), we measured synapse loss and toxicity surrounding the plaque types in hippocampal brain sections (Fig. 4C) using antibodies against LAMP1 and HOMER1, respectively. Higher levels of dystrophic neurites (i.e., LAMP1 levels, Fig. 4D) were measured around more structurally mature plaque types compared to their less aggregated counterparts (Fig. 4E, F). LAMP1 signal intensity continuously decreased with

increasing distance from the plaque core (Fig. 4E, F), demonstrating elevated neuritic dystrophy at the plaque center that gradually decreases in the periphery. Furthermore, synapse loss correlated with increased plaque aggregation, with Aβ + h + q+ plaques demonstrating the greatest reduction in HOMER1 signal (Fig. 4G, H). HOMER1 signal reductions were most pronounced at the plaque center, with only minimal decreases being detected at a distance of 10 μm and no signal reduction being observed beyond 10 μm (Fig. 4G, H).To evaluate whether synapse loss is linked to increasing plaque age, as indicated by our correlative MALDI MSI-GeoMx analysis (Fig. 3H, N), we leveraged the previously established association between plaque age and elevated q+ staining (Fig. 2F). Specifically, we correlated HOMER1 signal intensity with the hyperspectral q/h ratio in individual 18-month-old mice (Fig. 4I). In line with our previous findings, HOMER1 intensity was negatively correlated with the LCO q/h hyperspectral ratio, supporting a relationship between synapse loss and plaque age.

## Discussion

Most studies attempting to understand the development of AD and neurodegeneration primarily consider factors such as chronological age and pathological severity. However, a detailed understanding of AD development still lacks a single plaque evolution-related perspective. For instance, it is unclear whether newly formed plaques exert a different influence on surrounding tissues compared to older plaques in the same individual. Furthermore, it remains an important question in AD research why Aβ plaques have a poor correlation with cognitive decline despite the amyloid hypothesis suggesting it as the disease trigger[33]. This gap in understanding is further emphasized by a population of people who live dementia-free lives but upon post-mortem examination have a brain populated with plaques (cognitively unimpaired, amyloid positive CU-AP individuals)[34,35]. Interestingly, the brains of these patients are characterized by an abundance of diffuse plaques, suggested to be an immature state that exerts little toxicity to its surroundings. These findings suggest a close relationship between plaque age, structural maturity, and neurotoxicity. In this study, using a multiomic imaging of stable isotope labeling kinetics (iSILK) driven approach, we demonstrate this by showing that in older mice (18-month-old), older plaques are associated with higher structural maturity than newly formed plaques, which is in turn associated with greater neurotoxicity and more extensive loss of synapses.

Traditional bulk sequencing methods require mouse models of Alzheimer's disease to develop exaggerated levels of plaque pathology in order to minimize non-pathological tissue to capture plaque-induced transcriptomic changes[36]. Therefore, more disease-relevant Alzheimer's disease models such as $App^{NL-F}$ mice, that develop plaques gradually, with onset in older ages and without APP overproduction, were unable to produce sufficient plaque loads to influence bulk sequencing results (Supplementary Fig. 11). While our previous work, along with other studies, has leveraged recent advancements in spatial

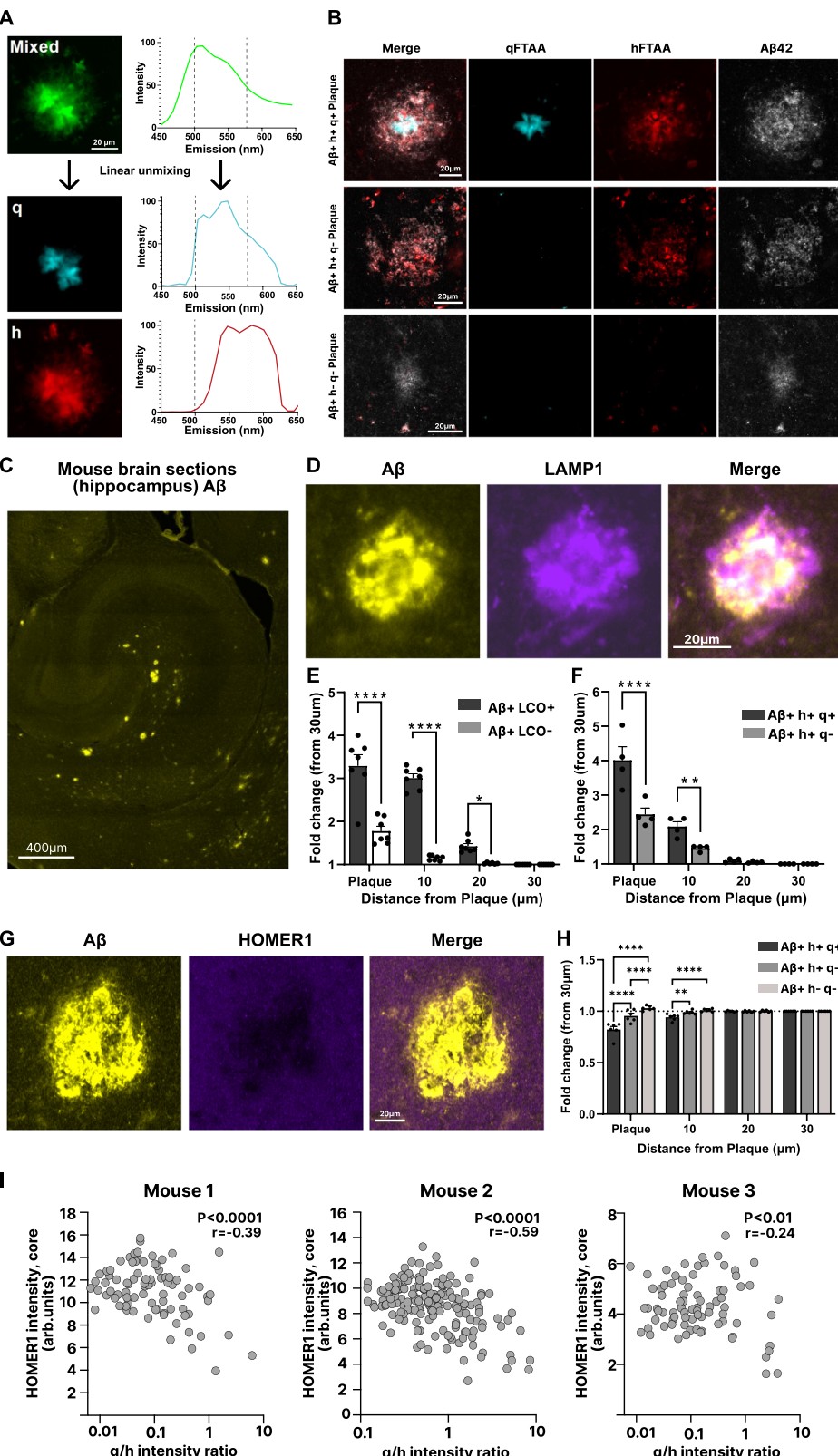

transcriptomics to demonstrate plaque-specific alterations in gene expression[32,36–39], these studies often combine plaque regions and therefore do not consider the heterogeneity of plaque types and maturity. This raises questions about the differential effects of plaque heterogeneity on surrounding tissue. Therefore, we have integrated spatial transcriptomics with a metabolic labeling and spatial mass spectrometry imaging paradigm (iSILK MALDI MSI) to provide insights into how gene expression evolves in conjunction with plaque maturation at the single plaque level.

We observed a negative correlation between synapse-associated genes and plaque age in mice at both 10 months (Experimental Design 1) and 18 months (Experimental Design 2) of age, indicating that while the loss of synaptic function is associated with increasing plaque age, this relationship stands at both early and late chronological stages. Notably,

**Fig. 4 | Differences in Synapse Loss and Toxicity Revealed by LCO-Defined Plaque Types. A** Schematic illustrating the linear unmixing of mixed LCO signal into isolated q-FTAA and h-FTAA channels. LCOs are conformation sensitive fluorescent dyes, bind to amyloid structures. **B** Representative images of three LCO defined plaque types, showing q-FTAA, h-FTAA, and Aβ42 immunoreactivity. **C** Overview image of a hippocampal mouse brain section, stained for Aβ (**D**) Representative image showing increased LAMP1 expression colocalizing with plaques. **E, F** Quantification of LAMP1 protein fluorescence intensity at increasing distances from different LCO-defined plaque types. Two-way repeated measures ANOVA (E: $n = 7$ (2 m + 5 f) mice, plaque; $p = 1.6 \times 10^{-9}$, 10 μm; $p = 5.8 \times 10^{-11}$, 20 μm; $p = 0.018$, F: $n = 4$ (1 m + 3 f) mice, plaque; $3.6 \times 10^{-6}$, 10 μm; $p = 0.0045$). **G** Representative image illustrating HOMER1 loss associated with plaques.

**H** Quantification of HOMER1 protein fluorescence intensity at increasing distances from different LCO-defined plaque types. Two-way repeated measures ANOVA ($n = 6$ (3 m + 3 f) mice, plaque: Aβ + h + q+ vs Aβ + h + q-; $p = 1.0 \times 10^{-10}$, Aβ + h + q+ vs Aβ + h-q-; $p = 1.0 \times 10^{-15}$, Aβ + h + q- vs Aβ + h-q-; $p = 8.4 \times 10^{-6}$ & 10 μm: Aβ + h + q+ vs Aβ + h + q-; $p = 0.0026$, Aβ + h + q+ vs Aβ + h-q-; $p = 3.5 \times 10^{-5}$). **I** Pearson correlation of HOMER1 immunofluorescence intensity with q/h-FTAA intensity ratio, stratified by individual 18-month-old mice (mouse 1:m, mouse2:f, mouse3:f). Data are presented as mean ± SEM. Post-hoc Tukey's test corrections applied. Significance levels: ****$P < 0.0001$; ***$P < 0.001$; **$P < 0.01$; *$P < 0.05$. Source data are provided as a Source Data file. **B, C** Representative images selected from a dataset of 2407 plaques across 16 hippocampal sections from $n = 6$ 18-month $App^{NL-F}$ mice, with similar results observed throughout each biological replicate.

this is in line with SILK proteomic experiments in mice showing impaired turnover of synaptic proteins[8]. Thus, plaques are increasingly toxic to synapses as a result of continued presence in the tissue, which is independent of the age of the animal. Whereas findings from other impactful spatial transcriptomics studies on AD brain tissue[37,39] have also supported accumulated rather than acute plaque toxicity, they were unable to directly link plaque age with increasing synaptic dysfunction, entirely decoupled from chronological mouse age. In fact, many studies have focused on the spatial rather than temporal aspect of plaque toxicity as well as focused on subsets of genes associated with e.g., glial activation in response to plaque pathology[32,40]. Our study, in contrast, allowed us to evaluate the synaptotoxic effects of maturing plaques independent of mouse age on a whole transcriptomic scale, with key findings being validated with direct immunological staining for HOMER1 and LAMP1. Additionally, the mouse model used in this study more accurately reflects human AD as it combines WT Aβ pathology together with ageing, which contrasts with the $App^{NL-G-F}$ mouse model, on which most prior conclusions have been based.

Specifically, in both 10 and 18-month-old mice, downregulation of synaptic genes primarily affected processes governing excitatory synaptic structure, function, and plasticity. Enriched GO terms included postsynaptic organization, synapse assembly regulation, dendritic spine development, and cognitive processes like learning and memory. Most impacted genes, such as *Syngap1, Dlg4, Nptx2, Dlgap1, Grin2a, Bdnf*, and *Camk2a*, encode proteins crucial for maintaining and dynamically remodeling glutamatergic synapses, especially in dendritic spines[41–43]. This selective vulnerability highlights excitatory synapses as central in the earliest molecular events associated with amyloid-induced synaptic dysfunction, preceding significant neuronal loss, and aligns with the widely held view that glutamatergic synaptic impairment is a fundamental early hallmark of AD progression[44,45]. The negative correlations with plaque age can be a result of functional synaptic impairment or neuronal degradation. Preferably, one would like to understand if e.g., neuropil translocated genes are differentially affected with plaque age as compared to neuronal somatic genes as this would speak to a functional impairment and degradation of synapses. This is, however, hard to delineate as the GeoMx approach used here did not capture gene expression on a single cell- or even subcellular level. The present data did not show any difference in cell count to be associated with plaque age, however, a plaque age-dependent decrease was observed with IHC of the post synaptic marker HOMER1. While both suggest that, in fact, plaque age is associated with synapse-specific impairment relative to overall neuronal degradation, this constitutes a very relevant aspect to be investigated in follow-up studies.

In addition to the synaptic changes, in 10-month-old mice, we captured an initial immune response and increasing immune proliferation in response to the very first plaques. Surprisingly, this increase in immune response-associated genes with increasing plaque age was not reflected in our GO analysis results of 18-month-old mice[46]. However, this does not necessarily indicate a diminished immunological response but may rather reflect that immune-related genes are

equally upregulated across plaque ages in older animals because of the heavier plaque load[32]. In contrast, at 18 months, we observed an increase in metabolic processes-associated genes with increasing plaque load. This potentially reveals that increasing toxicity of older plaques results in increasing metabolic demand on the surrounding cells or possibly toxicity leading to mitochondrial dysfunction and resulting compensation[41–43,47–49].

Although changes in synaptic, immune, and metabolic gene expression could reflect shifts in the cellular composition surrounding plaques, no significant correlation was observed between the GFAP immunological staining intensity and plaque age, suggesting that the number of astrocytes surrounding plaques did not systematically vary with plaque maturity. Since GFAP is generally regarded as a marker of proliferation rather than activation, this lack of correlation does not necessarily imply an absence of astrocytic reactivity to plaques. Indeed, we observed strong positive correlations of several DAA genes (*S100a10, Cxcl5, Pdlim4*)[31,50] with increasing plaque age in both 10- and 18-month-old mice. This is in line with the recently reported role of astrocytes in Aβ plaque formation and synaptic loss, in which astrocytic cross-talk with both neurons and microglia intensifies in tandem with increasing plaque maturity[39]. Aligned with these findings, we also observed the upregulation of several DAM-related genes (e.g., *Axl, Anxa1, Ctsd, Csf1*) with plaque maturation in both 10- and 18-month-old mice[51–56], underlining that, as mentioned above, the immunological response to plaques likely extends beyond the earliest stages of plaque formation. In 10-month-old mice, increased expression of genes associated with T cell activation (e.g., *H2-K1, Il6, Irf1*) and IFN-γ−responsive microglial genes (e.g., *Anxa1, Socs5*) reflects a combined adaptive and innate immune response during plaque maturation[51–53,55]. Such immune activation is likely to contribute to progressive myelin and oligodendrocyte damage, in line with recent mechanistic models of neurodegeneration[51,57]. Despite significant enrichment of T cell−associated ontologies, the driving genes are also broadly expressed in microglia, while canonical T cell markers (*CD3, CD4, CD8*) were not significantly changed. Therefore, potential T cell infiltration cannot be reliably distinguished from microglial activation in this dataset.

Recent advances in amyloid imaging have introduced structural-specific dyes, facilitating the easy categorization of plaque types through hyperspectral imaging and linear unmixing. This method offers an unbiased, more efficient and high-throughput alternative to previous techniques that relied on morphological examination by pathologists. We have identified three distinct populations of amyloid plaques in $App^{NL-F}$ mice, characterized by their unique structural isoforms. These populations were: Aβ + h + q +, Aβ + h + q-, and Aβ + h-q-, likely corresponding to the morphologically defined dense cored, fibrillar, and diffuse plaques respectively[58]. Our findings indicate that in older $App^{NL-F}$ mice, the majority of deposits exhibit a diffuse and non-pleated structure (Aβ + h-q- / diffuse populations), which induces lower LAMP1 levels and HOMER1 loss compared to more compact and larger LCO positive plaques. Similarly, HOMER1 was found to negatively correlate with plaque structural maturity and age, respectively. This finding of increased synapse loss with increasing plaque age and

structural maturity is in line with previous findings in humans on morphologically- and thioflavin-defined plaques[22,59,60] and also aligns with findings in amyloid-positive CU-AP individuals, where diffuse plaques exhibit low levels of neurotoxicity[61,62]. Furthermore, the finding of increased neurotoxicity with increasing aggregation is also consistent with studies showing that a higher propensity for Aβ aggregation and the presence of dense cored plaques are associated with increased progression of clinical dementia[35,63,64].

The process of precipitating plaque pathology is difficult to capture with conventional, steady state imaging techniques, particularly in comparably late-onset mouse models like $App^{NL-F}$. Dynamic imaging approaches such as chemical or genetic in vivo labeling provided important insight, though, despite being in vivo, lack chemical specificity and/or temporal span[46]. Using the iSILK-based approach employed here for dynamic imaging of amyloid peptide deposition, our results from 10-month-old (Experimental Design 1) $App^{NL-F}$ mice indicate that a plaque center forms first as a small dense core which is however less fibrilized at the beginning and gradually compacted upon plaque maturation. We have seen similar results in a previous iSILK study on the $App^{NL-G-F}$ model suggesting that plaques initially precipitate as small dense deposits[21]. Indeed, the results show that plaques continue to fibrilize as we see a positive correlation between q-FTAA binding and increasing plaque age. This indicates a dynamic maturation process whereby pre-fibrillar Aβ (h-FTAA + ) species progress to a more mature state at the center of a plaque. This is of interest as a chronological, static imaging study on plaque maturation in $App^{NL-F}$ showed no difference in q/h emission across age, though a large variety of plaque maturation stages was observed across plaques within an age group[28]. Our results thus provide a first indication that the LCO q/h emission ratio could be a useful proxy to determine plaque age in future work, without the requirement of time-consuming and costly iSILK experiments.

A limitation of the current study is that, due to technical constraints[65], the MALDI-LCO as well as MALDI-GeoMx analyses had to be carried out on sequential tissue sections (Fig. 1D, E). This made it challenging to capture small pathological alterations spanning two consecutive sections, limiting the total number of plaques per animal and thereby reducing statistical power. To mitigate this issue, we conducted a validation experiment with a larger number of mice, leveraging the correlation between plaque age and hyperspectral ratio, to confirm our main findings pertaining to e.g., synapse loss correlation with plaque age. Another limitation includes the lack of cell-type-specific transcriptomic data on the single-plaque level. This is attributed to the limited number of morphological markers and sensitivity of GeoMx analysis. While imaging-based spatial transcriptomics methods such as e.g., CosMx (Bruker/Nanostring) or Xenium (10x) offer higher spatial resolution of transcripts near plaques, with CosMx now enabling near whole-transcriptome profiling, these approaches exhibit reduced gene-wise sensitivity compared to sequencing-based methods, limiting the reliable detection of lowly expressed genes and compromising quantitative accuracy.

Despite those limitations, using the current correlative spatial biology setup with with consecutive tissue sections would of course allow to interface iSILK/MALDI-MSI with other high resolution spatial biology techniques such as Merscope (Vizgen), CosMx, Vizium/ Xenium (10x Genomics), Codex (Akoya Biosciences) and imaging mass cytometry (MibiTof, Standard Bio Tools). Given the additional spatial resolution and biological information (e.g., cell specificity, proteins) expanding towards these tools constitutes an important goal for future studies.

Finally, the $App^{NL-F}$ mouse model, though advantageous for capturing age-related pathological changes, primarily shows Aβ1-42 deposition without major truncated or post-translationally modified Aβ isoforms as observed in humans – a common limitation of genetic mouse models for APP/AD pathology.

Together, we present a conceptual and technical innovative study using a spatial biology paradigm to detail the earliest events of plaque deposition through plaque maturation. This suggests that these plaques continuously fibrilize with age and that the oldest, most fibrilized plaques show the highest level of toxicity towards associated synapses, which is an accumulated phenomena due to a consequence of continuous presence, growth and interaction with synapses over time. The spatial and temporal resolution along with chemical precision achieved with the iSILK guided spatial biology approach, particularly with respect to the age of the AD mice studied, exceeds steady state technologies and consequently provides a dynamic dimension that would otherwise not be discernible.

## Methods

All procedures involving mice were performed at University College London following approval by the University College London Animal Welfare and Ethics Review Board (approved 06/05/2016) and in accordance with the UK Animals (Scientific Procedures) Act 1986. All experiments also adhered to guidelines of the Institutional Animal Care and Use Committee (IACUC) and relevant local ethical regulations.

### Animal model

Male and Female APP knock-in mice ($APP^{NL-F}$, developed and gifted by the Saido group at the RIKEN Center for Brain Science, Japan) carrying humanized Aβ sequence, along with the Swedish mutation (KM670/ 671NL) on exon 16 and the Beyreuther/Iberian mutations (I716F) on exon 17 were used in the study[23]. Transnetyx (Cordova, TN, USA) genotyping services were used to determine the presence of the knock-in genes in breeders. The mice used in this study were group-housed (2–5) with same sex littermates with an ad libitum supply of food and water, under a 12-hr light/dark photoperiod, at a controlled temperature and humidity.

### Experimental design

For all iSILK/MALDI-MSI experiments (Fig. 1D, E), $App^{NL-F}$ mice ($n = 7$) were isotopically labelled with $^{15}$N spirulina diet from age 6–10 months (PULSE period) and either culled immediately at 10 months old ($n = 3$), or after additional 8-month CHASE (washout) period at 18 months of age ($n = 4$). Each mouse required 1 g labelled diet per day. With 4 months labeling (ca 122 days i.e., 122 g/mice, $n = 7$) requiring 1 kg of labelled feed for all 7 mice. For experiments evaluating LCO-defined plaque types and their correlation with IHC stainings (Fig. 1F), non-metabolically labelled $App^{NL-F}$ mice (n = 18) were sacrificed at either 9-month ($n = 7$), 14-month ($n = 5$) or 18-month ($n = 6$) of age.

### Tissue collection

Animals were decapitated with the brain rapidly extracted on ice. For the metabolically labelled mice, one hemisphere was snap-frozen directly after isolation using liquid nitrogen ( − 150 °C)-cooled isopentane. For non-metabolically labelled mice, one hemisphere was snap-frozen directly after isolation on dry ice. In both cases the other hemisphere was drop fixed in 10% formalin (4% PFA) at 4 °C overnight. Formalin was subsequently washed out and replaced with 30% sucrose, 0.02% sodium azide in phosphate-buffered saline (PBS) solution for long term storage at 4 °C.

### Tissue Sectioning

For correlative GeoMx/MSI experiments, consecutive, sagittal cryosections were collected at 10 μm from fresh frozen brain tissue on a cryostat microtome (Leica CM 1520, Leica Biosystems, Nussloch, Germany) at −18 °C. Sections were thaw mounted on SuperFrost Plus Slides for the GeoMx DSP platform, with consecutive sections mounted on conductive indium tin oxide (ITO) glass slides (Bruker Daltonics, Bremen, Germany) for MALDI-MSI. All tissue was stored at −80 °C.

PFA-fixed brain tissue, collected for plaque staining and immunohistochemistry, was sectioned transverse to the long axis of the hippocampus at 30 μm using a frozen sledge microtome (Leica). Free-floating sections were collected and stored in 0.02% sodium azide in PBS at 4 °C.

## Matrix deposition for MALDI-MSI

Fresh frozen tissue sections on ITO-coated glass slides were fixed in 100% ethanol for 60 seconds, followed by 70% ethanol for 30 seconds. Lipids were removed by immersing the sections in Carnoy's solution (6:3:1 ethanol/chloroform/acetic acid) for 110 seconds, followed by subsequent washes in 100% ethanol for 15 seconds, 0.2% trifluoroacetic acid in water for 60 seconds, and 100% ethanol for 15 seconds. The tissues then underwent peptide retrieval using formic acid vapor for 20 minutes. To facilitate peptide ionization and desorption, a matrix solution (15 mg/ml 2,5-dihydroxyacetophenone in 70% acetonitrile, 2% acetic acid, 2% trifluoroacetic acid) was applied using a TM sprayer (HTX Technologies, NC, USA) and recrystallized in 5% methanol vapor at 85 °C for 3 minutes.

## MALDI-MSI data acquisition

MALDI-MSI was performed using a MALDI-time-of-flight (TOF) instrument (RapifleX, Bruker Daltonics) equipped with a scanning Smartbeam 3D laser. The acquisition was conducted at a pixel size of 10 μm with 200 shots per pixel at 90% power with a shot frequency of 10 kHz. Spectra were collected within the mass range of 1500–6000 m/z in linear positive mode due to higher sensitivity at high spatial resolution for correlative spatial transcriptomics (mass resolution: m/Δm = 1000 (FWHM) at m/z 4515). A second set of data for correlative LCO microscopy was collected at 20 μm with 200 shots per pixel at 90% power with a shot frequency of 10 kHz in reflector mode (mass resolution: m/Δm = 15000 (FWHM) at m/z 4512.3). The system was externally calibrated by spotting a mixture of peptide standard II and protein standard I (Bruker Daltonics).

## Spatial transcriptomics

**Slide preparation.** A number of 3 (glass 1) and 4 tissue sections (glass 2), respectively were mounted on in total two glass slides as per the manufacturer's recommendations. Slides were processed following the standardized protocol for fresh frozen samples for analysis at the GeoMx Digital Spatial Profiler (provided for customers at Nanostring University). The 10 μm cryosections were fixed in 10% formalin followed by ascending concentrations of EtOH. Slides were then immersed in 1X Tris EDTA at 100 °C for 15 minutes in a pressure cooker. After washing in PBS, RNA targets were exposed to 1 ug/μl of Proteinase K in PBS at 37 °C for 15 minutes. Tissues were post-fixed in 10% formalin for 5 minutes at RT followed by 2 × 5-minute washes in NBF Stop Buffer at RT. Tissues were subsequently incubated in whole transcriptome RNA probes overnight in a hybridization chamber held at 37 °C. Off-target probes were washed out in 2X Stringent Solution or 25 minutes at RT. Nonspecific binding was blocked using Buffer W (Nanostring proprietary blocking buffer) followed by incubation with morphology markers, conjugated antibodies in Buffer W for 2 hours at RT (mouse anti-GFAP Alexa-Fluor 488 conjugate (1:500, Invitrogen, #53-9892-82), mouse anti- Aβ40/42 Alexa Fluor 594 conjugate (1:500, Nanostring, #121301306). After which, nuclei were counterstained for 1 hour at RT (SYTO 13, #S7575). After a final wash in 2x SSC buffer, slides were placed into the GeoMx instrument.

**AOI selection.** MALDI MSI images of consecutive sections were overlayed on the fluorescent image output from the GeoMx. Plaques spanning both sections were identified, and a circular area of illumination (AOI) three times the radius of the plaque was centered around the plaque's midpoint. All probes regardless of cell type were extracted per AOI.

**Library preparation and readout.** DNA barcode oligomers were aspirated into well plates whereby the aspirate from one plaque corresponds to one well. The well plates were then dried and rehydrated in 10 μl nuclease-free water and PCR cycled with GeoMx SeqCode master mix and SeqCode primers (Nanostring Technologies). Finally, the PCR product was purified with AMPure beads. Plates were subsequently sequenced on an Illumina NextSeq2000. FASTQ files were converted to *.dcc format with the GeoMx NGS Pipeline GUI.

## Immunohistochemistry and LCO staining

Free-floating 30 μm sections underwent antigen retrieval in 10 mM pH 9 Tris-EDTA for 30 minutes at 80 °C. Subsequently, sections were permeabilized in 0.3%Triton X-100 in PBS (PBST) for 3 × 10 minutes at RT before non-specific binding was blocked in 3% goat serum in PBST (Blocking solution) for 1 hour at RT. Sections were incubated with primary antibodies diluted in blocking serum at 4 °C overnight (1:500 mouse anti-Aβ (6E10, BioLegend, #SIG-39320), 1:500 rabbit anti-Aβ (Thermo Fisher Scientific, #700254), 1:200 chicken anti-HOMER1 (Synaptic Systems, #160001), 1:500 rat anti-LAMP1 (Abcam, #ab25245)). Sections were washed in PBST 3* 10 minutes at RT followed by incubation in secondary antibodies diluted in blocking solution for 2 hours at RT (1:500 goat anti-mouse AF594 (Thermo Fisher Scientific, #A11032), 1:500 goat anti-rabbit AF594 (Thermo Fisher Scientific, #A11037), 1:500 goat anti-chicken AF647 (Thermo Fisher Scientific, #A-21449), 1:500 Donkey anti-rat AF594 (Thermo Fisher Scientific, #A-21209)). Nuclei were counterstained with 4′,6-diamidino-2- phenylindole (Abcam, #ab228549) for 5 minutes at RT. For structural amyloid analysis, tissues were stained with two luminescent conjugated oligothiophene (LCO) dyes, tetra- (q) and heptameric (h-) formyl thiophene acid (FTAA) obtained as a kind gift from Prof Peter Nilsson at Linköping University, Sweden where they were developed. For linear unmixing analysis after antibody staining, sections were subsequently incubated for 30 minutes at RT with 3 μM q-FTAA and 1μM h-FTAA. For LCO MALDI correlation analysis, cryo-sections were fixed sequentially in a gradient concentration of ice-cold 95%, 70% ethanol, and 1× phosphate-buffered saline (PBS) at room temperature followed by incubations in q-FTAA, 2.4 μM in Milli-Q water and h-FTAA, 0.77 μM in Milli-Q water) in the dark for 25 minutes.

## Fluorescent microscopy

All imaging was performed on a Zeiss LSM880 confocal microscope. The imaging of double LCO-stained tissues was performed in hyperspectral mode using the 32-channel GaAsP spectral detector. The dyes were stimulated using a 35 nW, 458 nm argon laser. For LCO correlation analysis, continuous emission spectra were obtained in lambda mode. For plaque-type classification, linear unmixing mode utilized reference spectra of q-FTAA and h-FTAA to isolate the corresponding signals as individual channels.

For LCO correlation analysis individual amyloid plaques were randomly selected and images as z-stacks (20 stacks per plaque, 1-μm apart) under a 20X air objective. The 500/580 nm emission ratios were collected using Zen Black 2.3 software.

In the linear unmixing analysis, LAMP1, Aβ, HOMER1, q-FTAA and h-FTAA images were taken as z-stacked (3 μm apart) tile scans of whole hippocampus using a 20X air objective. LAMP1, Aβ and HOMER1 were imaged in channel mode using constant light, gain and exposure settings matched to the fluorophore.

## Ex situ Aβ peptide analysis

**LC-MS/MS proteomic analysis of App^{NL-F} mouse brain tissue.** To confirm the presence of Aβ1-42 as the predominantly produced peptide species in *App^{NL-F}* mice and investigate potential post-translation modifications, we performed two separate LC-MS/MS proteomic analyses of 1) purified amyloid fibrils and 2) immunoprecipitated Aβ peptides, as detailed below.

**Purification of amyloid fibrils from App^{NL-F} mouse brain tissue.** We performed discovery-based bioinformatic analyses of an existing high-

resolution tandem mass spectrometry (MS/MS) dataset obtained in the Savas lab at Northwestern University in Chicago, USA. This MS/MS data was generated from tryptic digests of purified ($n = 5$) and highly purified amyloid fibrils ($n = 8$), isolated using a recently developed biochemical purification strategy (i.e., detergent solubilization, differential centrifugation, and ultra-sonication) for $App^{NL-F}$ mouse brain extracts[66]. Mouse brain tissue samples (0.25–1 g) were homogenized in 1 ml of an extraction buffer composed of 10 mM Tris–HCl (pH 7.0), 0.25 M sucrose, 3 mM EDTA, 0.1% sodium azide, and a protease inhibitor cocktail. Homogenates were incubated overnight at 4 °C with gentle end-over-end agitation. The next day, dry sucrose was added to increase the final concentration to 1.2 M, and samples were subjected to ultracentrifugation at 250,000 × g for 45 minutes at 4 °C. After carefully removing the upper and intermediate layers, the remaining pellet was resuspended in the same buffer adjusted to 1.9 M sucrose and centrifuged again at 125,000 × g for 30 minutes.

Following centrifugation, the pellet was washed twice with 1 ml of 50 mM Tris–HCl buffer by spinning at 8000 × g for 15 minutes at 4 °C. To remove residual cellular material, the pellet was incubated at 37 °C for 3–4 hours in a digestion solution containing DNase I and collagenase, then washed again with Tris–HCl. Subsequently, the sample was solubilized in Buffer A supplemented with 1.3 M sucrose and 1% SDS and centrifuged for 1 hour at 200,000 × g. The pellet from this step was retained and the supernatant was diluted to reduce the sucrose concentration to approximately 1 M and centrifuged again at 250,000 × g for 45 minutes. Both resulting pellets were pooled and dissolved in 200 µl of Tris buffer. The enriched amyloid-containing fraction was then sonicated using water bath sonication (Bioruptor Pico Plus, 15 cycles, medium setting), followed by five sequential washes with 1% SDS in Tris buffer (16,000 × g, 20 minutes, 4 °C). Proteins were precipitated using the chloroform/methanol method, then denatured in 50 µL of 8 M urea prepared in 50 mM ammonium bicarbonate (ABC). An equal volume of 0.2% ProteaseMAX (Promega, V2072) in ABC was added, and samples were vortexed for 1 hour. Disulfide bonds were reduced with 5 mM TCEP for 20 minutes at room temperature and alkylated with 10 mM iodoacetamide in the dark for 15 minutes, followed by quenching with 25 mM TCEP. Proteins were digested overnight at 37 °C with MS-grade trypsin (Promega, V5280). Digestion was halted the next morning with 1% formic acid. Peptides were desalted using C18 spin columns (Thermo Scientific, 89870) according to the manufacturer's protocol, dried in a refrigerated speed vacuum, and stored at −80 °C.

**Liquid chromatography–tandem mass spectrometry (LC-MS/MS) analysis of purified amyloid fibrils.** Dried peptides were resuspended in 20 µL of buffer (94.875% H₂O, 5% acetonitrile, 0.125% formic acid). A 3 µg aliquot, quantified by micro-BCA assay (Thermo Fisher Scientific, 23235), was injected using either a Thermo EASY nLC 100 UPLC or UltiMate 3000 HPLC system. Samples were loaded onto a Pepmap 100 trap column (75 µm × 2 cm) coupled to a nanoViper analytical column, with nanospray ionization applied at 2000 V. Data acquisition was performed on an Orbitrap Fusion tribrid mass spectrometer (Thermo Fisher) using label-free quantification settings: 300 °C ion transfer tube, 60 K resolution, 300–1500 m/z scan range, 3 s cycle time, AGC target of 200,000, and dynamic exclusion (10 ppm, 30–45 s). Collision-induced dissociation (CID) used the ion trap detector with rapid scan and 75 ms injection time. For absolute quantification of Aβ42 peptides, targeted MS/MS runs were performed using a predefined m/z list.

Raw MS files were processed using the offline RawConverter tool (http://fields.scripps.edu/rawconv/) to generate MS1 and MS2 data files. These were then analyzed on the Integrated Proteomics Pipeline (IP2, Bruker: http://www.integratedproteomics.com/), a web-based platform for peptide identification and quantification.

Obtained MS/MS spectra were subjected to ProLuCID protein database searches. ProLuCID is an improved SEQUEST-like algorithm with enhanced sensitivity and specificity[67]. Two independent searches were performed against the complete mouse reference protein database from UniProt, supplemented with an additional entry for the amyloid precursor protein (App) containing a humanized Aβ sequence, as engineered in the Saito NL-F mouse model. The searches included differential modifications for phosphorylation (+ 79.966331 Da on serine, threonine, or tyrosine [STY]) and pyroglutamate formation (−18.0106 Da from glutamate or −17.0265 Da from glutamine)[68]. Searches allowed precursor mass tolerances of 20 ppm and fragment ion tolerances of 600 ppm, including fully and half-tryptic peptides without restricting missed cleavages. Peptides were assembled and filtered with DTASelect2 (v2.1.3), applying static carbamidomethylation (+ 57.02146 Da) on cysteines. Peptide confidence and false discovery rates (FDR) were controlled using a target-decoy approach, setting a minimum peptide length of five amino acids and maintaining a 1% FDR at the protein level.

**Immunoprecipitation of Aβ peptides in App^NL-F mouse brain extracts.** Fresh frozen $App^{NL-F}$ cerebral tissue (90–110 mg pieces, $n = 2$) was homogenized in 600 µl tris(hydroxymethyl)aminomethane (Tris)-buffered saline (TBS), pH 7.6, using one 5 mm bead per sample in a TissueLyser (Qiagen) for 4 min at 30 Hz. To the homogenate(~700 µl), 400 µl TBS was further added and transferred to a new tube and centrifuged at 31,000 g for 1 h at +4°C. The supernatant (TBS fraction) was transferred to a new tube and stored at −80°C until further use. The pellet was resuspended in 1 ml of 70% FA (v/v), followed by further homogenization in the TissueLyser for 2 min at 30 Hz and subsequent sonication for 30 s. The homogenate was centrifuged again at 31,000 g for 1 h at +4°C and the supernatant (FA fraction) was dried down in a vacuum centrifuge.

Prior to IP, the dried FA fractions were reconstituted in 200 µl 70% FA (v/v), shaken for 30 min at room temperature 21 °C, and centrifuged at 31,000 g for 1 h at +4°C. The supernatant was removed and neutralized with 4 ml 0.5 M Tris.

IP was performed with a KingFisher (Thermo Fisher Scientific) magnetic particle processor[69]. Briefly, 4 µg of antibodies 6E10 and 4G8 (BioLegend) were separately added to 25 µL of Dynabeads M-280 (Thermo Fisher Scientific) sheep anti-mouse suspension, according to the manufacturer's product description. The antibody-bead complexes were combined (50 µL/sample in total) and added to all samples together with 20% (v/v) Triton X-100 to a final concentration of 0.2% (v/v) and incubated overnight on a Rock'n'Roller at +4 °C. FA fraction and LCM samples were neutralized to pH 7.0 before the beads were added. The beads/FA fraction was transferred to the KingFisher for automatic washing (in 0.2% Triton X-100, phosphate buffered saline (PBS), pH 7.6, and 50 mM ammoniumbiocarbonate) and elution in 0.5% FA. The eluate was dried down in a vacuum centrifuge pending MS analysis.

**Liquid chromatography–tandem mass spectrometry (LC-MS/MS) analysis of immunoprecipitated Aβ peptides.** To analyze the immunoprecipitated $Aβ$ peptides, a set-up consisting of nanoflow liquid chromatography (LC) coupled to an electrospray ionization (ESI) hybrid quadrupole–orbitrap tandem mass spectrometer (Dionex Ultimate 3000 system and Q Exactive, both Thermo Fisher Scientific) was employed[70,71]. Samples reconstituted in 7 µL 8% FA/8% acetonitrile in water (v/v/v) were loaded onto an Acclaim PepMap 100 C18 trap column (length: 20 mm; inner diameter: 7 µm; particle size: 3 µm; pore size: 100 Å) for online desalting, and subsequently onto a reversed-phase Acclaim PepMap RSLC column (length: 150 mm, inner diameter: 75 µm; particle size: 2 µm; pore size: 100 Å) for separation (both Thermo Fisher Scientific). Mobile phases were 0.1% FA in water (v/v) (A) and 0.1% FA/84% acetonitrile in water (v/v) (B). Separation was performed at a flow rate of 300 nL/min using a linear gradient of 3% to 40% B for 50 min at 30 °C. The mass spectrometer was operated in

positive ion mode and set to acquire spectra between m/z 350 and 1,800. MS and MS/MS acquisitions were both obtained at a resolution setting of 70,000 using 1 microscan. MS/MS acquisitions were obtained using higher-energy collisional dissociation fragmentation (HCD) using a normalized collision energy (NCE) setting of 25, target values of 106, and maximum injection time of 250 ms. Database search was performed with PEAKS Studio v12.5 (Bioinformatics Solutions) against a custom-made APP database. Search parameters included precursor mass error tolerance: 20 ppm, fragment mass error tolerance: 0.05 Da, enzyme: none, digest mode: unspecific, variable modifications: Deamidation (NQ), Oxidation (M), Phosphorylation (STY), Pyro-glu from E, max. Variable PTMs per peptide: 2. All fragment mass spectra were also evaluated manually.

## MALDI-MSI data analysis

**Calculation of nitrogen index in linear positive (LP) mode**. We implemented a robust pipeline to detect, model, and characterize peaks from MALDI-MSI spectra, optimizing accuracy through an asymmetric Gaussian fitting approach for linear positive mode (LP). First, local maxima were identified by filtering for intensities higher than their immediate neighbors, followed by selecting the main peak based on maximum intensity. A fitting range, extending symmetrically around the main peak, was established to include relevant peak data while excluding noise. The Asymmetric Least Squares (ALS) baseline correction was applied, and parameters were set up with a lambda = 5, asymmetry weight ($p = 0.025$), and maximum iterations (maxit = 100) to fit a baseline that accurately distinguished signal from background, with negative values set to zero. Initial parameters for the asymmetric Gaussian model, including peak amplitude, center (m/z value), and asymmetric widths for the left and right flanks of the peak, were estimated directly from the data. Model fitting was performed using a nonlinear least squares method with bounded constraints to ensure realistic parameter estimation. The asymmetric Gaussian model accounted for natural irregularities in peak shapes, providing high fidelity in capturing peak characteristics. Key metrics such as the FWHM and also goodness of fit ($R^2 > 0.90$) was derived from the fitted curve, offering detailed insights into the peak shape and spread. Based on the peak analysis performed pipeline, the FWHM values for each plaque ROI at 10 and 18 months were extracted for correlation analysis with gene expression from GeoMX AOI.

## Calculation of Nitrogen Index in reflector positive (RP) mode

A custom R-based Shiny application was developed for preprocessing mass spectrometry data and performing isotopic pattern analysis in reflector mode MALDI MSI. The pipeline, featuring an interactive interface built with shiny, plotly, dplyr, signal packages and custom functions includes two modules: Preprocessing (signal enhancement) and PeaksCatcher (peak detection and characterization). The ratio of the 4th to the 3rd isotopologue peak areas of Aβ1-42 was used to assess label incorporation corresponding to the relative age of the plaque. A detailed description and pipeline can be found on: https://github.com/MaciejDulewiczGU/MaldiGeoMxSpatialTranscriptomicsIsotopeaker and https://maciejdulewiczgu.shinyapps.io/IsotopeakeR_Beta.

## MALDI MSI - GeoMx data analysis

The Quality control (QC) analysis for NGS and Biological Probe QC was conducted utilizing the default GeoMX DSP parameters (GeoMX NGS Pipeline 2.3.4). All further analyses were carried out in R [v4.3.1, R Core Team, Vienna Austria, (https://www.R-project.org/). Probes were excluded from analysis if counts were too low or failed the Grubbs outlier test according to NanoString guidelines. Nine AOIs in the 10-month-old group and six AOIs in the 18-month-old group were excluded from further analysis due to ambiguous spot morphology >;1 plaque or very low intensity in MALDI results (Fig. 3A). After data filtering, the geometric mean was calculated for targets in each AOI. QC

results were compared with those from the standard NanoString pipeline, showing generally consistent outcomes in outlier detection. Any minor discrepancies are likely attributable to slight differences in the test parameters. Quantile normalization was applied using the normalize: quantiles function provided by the *preprocessCore* R package. The full width at half maximum of each Aβ1-42 peak from the MALDI imaging results (LP mode) was used as an estimate for [15]N incorporation (Nitrogen index) and was calculated for the peak area outlined between the local intersection with the fitted baseline[21] (Fig. 1E, 3B). Pearson's correlation was run between quantile normalized gene expression and FWHM MALDI MSI data for Aβ1-42. The correlation results were divided into positive and negative, considering age (10-month-old positively/negatively and 18-month-old positively/negatively correlated). For data visualization in volcano plots (Fig. 3C, F) uncorrected (non-FDR-adjusted) $P < 0.05$ was used as significance cut-off. Detailed analysis results, both volcano plots and correlation scatterplots (Nitrogen Index LP vs. Quantile normalized gene expression) for individual genes, can be explored in the ShinyApp (link available in the Data Availability section). Gene over-representation enrichment (ORA) analysis was performed for each group separately, including the three different Gene Ontology (GO) terms related to Biological Processes (BP), Cellular Component (CC) and Molecular Functions (MF). For the ORA analysis, we applied a p-value and q-value cutoff of 0.05, using the entire gene set as a background. This analysis was conducted using the clusterProfiler R package[72]. Data visualizations were carried out using in-house functions and modification of already existing functions, as a part of packages: GOplot, StringDB, tidyr, msigdbr.

Image generated by GeoMx platform was used to quantify GFAP signal in each AOI. A uniform thresholding method was applied across all images to identify GFAP immuno-stained area, the analysis was performed using Qupath v0.5.1[73]. The GFAP-positive area in each AOI was then normalized to total AOI area. The normalized GFAP signal was correlated with *Gfap* expression levels (quantile normalized) and Nitrogen index (LP mode).

## LCO and MALDI MSI correlation analysis

For analyses comparing hippocampal and cortical plaques in mouse samples z-stacked images of individual plaques were analyzed in lambda mode, selecting the plane that showed the most pronounced 500 nm shift for further examination. Within this plane, a small circle, just a few pixels in diameter, was placed on the area of the plaque exhibiting the highest 500 nm shift, indicating the core. At this location, intensities at both 500 nm and 580 nm were recorded for analysis.

For the MALDI imaging data acquired in reflector mode, plaque regions of interest (ROIs) were annotated, and total ion current normalized average spectra of the annotated ROIs were exported as.csv files in flexImaging (v5.0). Hyperspectral images were unmixed using pure q/h-FTAA spectra, intensities were measured as a small rectangle at the core of plaque. Images were unmixed using ZEISS ZEN v3.9 and intensities were extracted using Qupath v0.5.1[73]. Correlation analysis was then conducted between the q/h signal ratio of the hyperspectral data and the 4th/3rd isotopic peak area ratio of the same plaque detected in the corresponding MALDI imaging data. 7-10 plaques per animal from 3 animals were selected for this correlation analysis.

## IHC protein intensity towards plaque type

Image analysis was carried out using a set of custom macros for ImageJ (v2.16). In short, the Z-stack tile scans were projected to a single plane based on standard deviation. Regions of interest (ROIs) were then drawn around each hippocampus to restrict further analysis to this area. Plaques were subsequently thresholded and categorized based on the positivity of both Aβ and LCO dyes. Radiating rings from each plaque were created at 10 μm increments extending up to 30 μm with

any overlapping areas removed from the analysis. Mean gray values for each protein of interest at each increment were averaged per plaque type for each hippocampal image. 2-3 hippocampal images were averaged for each animal. Due to technical difficulties, LAMP1 analysis was conducted in two stages: The first stage involved assessing LAMP1 in plaques that were both Aβ and LCO positive (specifically h-FTAA only) and in plaques that were Aβ positive but LCO-negative. The second stage focused on plaques positive for both h-FTAA and q-FTAA, as well as plaques positive for h-FTAA but negative for q-FTAA. In contrast, HOMER1 analysis with Aβ, h-FTAA, and q-FTAA staining, was completed in a single batch.

### Weighted gene co-expression network analysis (WGCNA)

Weighted gene co-expression networks were constructed separately for transcriptomic data from 10- and 18-month-old mice using the R package *WGCNA*[74] including only genes significantly correlated with plaque age. The optimal soft-thresholding power was selected based on the criterion that the scale-free topology fit index ($R^2$) approached saturation near 0.9, with mean and median connectivity below 100. Signed networks were built using the *WGCNA::blockwiseModules* function with the following parameters: soft threshold power = 18/19, deepSplit = 2, corType = bicor, minModuleSize = 15, mergeCutHeight = 0.2, pamRespectsDendro = TRUE, pamStage = TRUE, maxPOutliers $p < 0.05$, and reassignThreshold = 0.05. After module detection, module Eigengenes, representing the first principal component of each module, were calculated and correlated with plaque age using Spearman correlation. $P$-values were adjusted for multiple testing using the Benjamini–Hochberg procedure. Detailed information on each gene's module membership, along with GO annotations for each identified cluster, can be explored at (https://maciejdulewiczgu.shinyapps.io/MALDI_GEOMX_VOLCANO/) at tab WGCNA Analysis.

### Statistics & reproducibility

Experimental group sizes for the iSILK feeding experiment were based on previous work using the $App^{NL-G-F}$ mouse model (Michno et al., 2021); to account for potential age-related mortality, one additional animal was included in the 18-month group, resulting in $n = 4$ for 18-month-old and $n = 3$ for 10-month-old mice. Group size estimation for immunohistochemistry validation was based on an a priori power analysis using expected effect sizes from pilot data: $d = 2.5$ for LAMP1 and $d = 1.5$ for HOMER1. Assuming $\alpha = 0.05$ and power = 0.8, the analysis indicated minimum sample sizes of $n = 3$ and $n = 5$ animals, respectively. For the IHC validation, investigators were blinded to experimental group during image acquisition and analysis. Outlier data points ( > 2 SD from the mean) were excluded. Specific regions of interest (ROIs) were removed from GeoMx analyses if they showed ambiguous morphology (e.g., overlapping plaques) or insufficient MALDI MSI signal.

ROI selection for GeoMx and MALDI MSI was not blinded or randomized, as all plaques spanning two consecutive sections were required for cross-modality alignment. For MALDI MSI analyses of $^{15}N$ enrichment in the hippocampus and cortex, as well as center versus periphery comparisons, plaques were randomly selected from within each brain region. In contrast, all identifiable plaques within the hippocampus were included for immunohistochemistry validation, and selection was therefore not randomized. To minimize potential bias, all image acquisition and analysis were performed in random order.

Statistical analyses were performed using GraphPad Prism 9 and R Studio. The specific statistical test used is described in each figure legend. For comparisons involving paired measurements within plaques (e.g., center vs. periphery), linear mixed-effects models were used to account for intra-plaque replication and inter-mouse variability. Where appropriate, two-sided paired or unpaired $t$-tests and Mann–Whitney $U$ tests were used for group comparisons, and Pearson correlation was used for continuous variable associations. Post hoc analyses using Sidak's correction were applied only when a statistically

significant interaction was detected. In all statistical tests, biological replicates were considered to avoid pseudoreplication.

For transcriptomic correlation analyses, genome-wide comparisons across 19,963 genes were conducted using Pearson correlation coefficients. As the application of multiple testing correction at the gene level (e.g., Benjamini–Hochberg) would restrict detection to only near-perfect correlations ($r \approx 0.9$), correlation $P$-values were instead used to rank genes for Gene Ontology (GO) enrichment analysis, with correction applied at the pathway level (FDR < 0.05). Full correlation statistics for all genes can be explored at: https://maciejdulewiczgu.shinyapps.io/MALDI_GEOMX_VOLCANO/.

### Reporting summary

Further information on research design is available in the Nature Portfolio Reporting Summary linked to this article.

## Data availability

The GeoMx transcriptomics data generated in this study are available in the Zenodo: (https://doi.org/10.5281/zenodo.16676234). All raw and processed MALDI–MSI data from plaque ROIs are available via an interactive Shiny application at (https://maciejdulewiczgu.shinyapps.io/MALDI_GEOMX_VOLCANO/) and (https://doi.org/10.5281/zenodo.16676234). Under the 'SPECTRA VIEWER' tab, all MALDI spectra from 18-month and 10-month groups can be viewed and downloaded. These data can be reanalyzed using the IsotopeakeR framework (see Code availability). Under the 'VOLCANO PLOT' and 'GO ENRICHMENT TABLES' tabs, statistical comparisons and correlation analyses of MALDI–MSI and GeoMx transcriptomics are available. The LC-MS/MS data generated in this study have been deposited in the PRIDE database under accession code PXD060410 (IP-LC-MS/MS) and MSV000092311 (proteomics of purified amyloid fibrils). Source data are provided with this paper.

## Code availability

The complete code for MALDI–MSI data processing, spectral analysis, and spatial integration with GeoMx transcriptomics is available at GitHub: (https://github.com/MaciejDulewiczGU/MaldiGeoMxSpatialTranscriptomicsIsotopeaker). Shiny app (IsotopeakeR beta): (https://maciejdulewiczgu.shinyapps.io/IsotopeakeR_Beta). Instructions for reanalyzing spectra downloaded from the 'SPECTRA VIEWER' tab of the MALDI–GeoMx portal using IsotopeakeR are provided in the GitHub repository and (https://doi.org/10.5281/zenodo.16676234 and https://doi.org/10.5281/zenodo.16729863).

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

## Acknowledgements

We thank the staff at Center for Cellular Imaging (CCI), Core Facilities, The Sahlgrenska Academy, University of Gothenburg, for microscopy expertise. The LCO probes were originally developed at Linköping University, Sweden by Peter Nilsson and Prof. Per Hammarström, and donated as a kind gift. National Institutes of Health (NIH): R01 AG078796 (JH, JNS, FAE), R21 AG078538 (JH), & R21 AG080705 (JH, JNS). Swedish Research Council (VR): 2023-02796 (JH), 2019-02397 (JH, HZ), Rådsprofessur (HZ), 2022-01018 (HZ), 2023-00356 (HZ), 2017–00915 (KB), & 2022–00732 (KB). Swedish Alzheimer Foundation: AF-968238, AF-939767, & AF-994281 (JH); AF-930351, AF-939721, AF-968270, & AF-994551 (KB). Swedish Brain Foundation: FO2022-0311 (JH), FO2022-0270 (HZ), FO2017–0243 (KB), & ALZ2022–0006 (KB). Swedish State Support for Clinical Research (ALF): ALFGBG-2025-2027 (JH), ALFGBG-71320 (HZ), ALFGBG-715986 (KB), & ALFGBG-965240 (KB). European Union (EU): Horizon Europe No. 101053962 (HZ), Marie Skłodowska-Curie No. 860197 (MIRIADE) (HZ), European Partnership on Metrology NEuroBioStand #22HLT07 (HZ), JPND2021-00694 (HZ), & JPND2019–466–236 (KB). Alzheimer's Specific Foundations: Cure Alzheimer's Fund (FAE, HZ), Alzheimer Research UK (FAE), Alzheimer's Drug Discovery Foundation (ADDF) #201809-2016862 (HZ), AD Strategic Fund & Alzheimer's Association #ADSF-21-831376-C, #ADSF-21-831381-C, #ADSF-21-831377-C, & #ADSF-24-1284328-C (HZ), Alzheimer's Association Zenith Award ZEN-21–848495 (KB), & Grant SG-23–1038904 QC (KB). Other Foundations: Race Against Dementia/Rosetree Foundation Team Award (JH), Bluefield Project (HZ), Olav Thon Foundation (HZ), Erling-Persson Family Foundation (HZ), Familjen Rönströms Stiftelse (HZ, KB), Åhlén-Stiftelsen #233011 (JH), Magnus Bergvalls Stiftelse, Stiftelsen Gamla Tjänarinnor (JH, HZ), & Gun och Bertil Stohnes Stiftelse (JH). La Fondation Recherche Alzheimer (KB), & Kirsten and Freddy Johansen Foundation (KB). University College London Hospitals Biomedical Research Center (HZ).

## Author contributions

J.H., J.I.W., and F.A.E. conceptualized the project. Methodological design and optimization. were carried out by J.I.W., M.D., A.S., K.S., J.G., S.K., S.D., J.N.S., L.F., D.P., G.B., and J.H. Experiments were performed by J.I.W., M.D., A.S., K.S., J.G., S.K., H.B.H., L.F., D.P., G.B., and J.H. Visualizations were produced by J.I.W., M.D., A.S., K.S., J.G., and S.K. Funding was. acquired by J.H., F.A.E., and J.N.S. J.H. and F.A.E. supervised and directed the research. The initial manuscript draft was written by J.I.W. and J.H., with review and editing contributions. from J.I.W., M.D., K.B., H.Z., D.M.C., J.N.S., F.A.E., J.H., and S.W.

## Funding

## Competing interests

H.Z. has served at scientific advisory boards and/or as a consultant for Abbvie, Acumen, Alector, Alzinova, ALZPath, Amylyx, Annexon, Apellis, Artery Therapeutics, AZTherapies, Cognito Therapeutics, CogRx, Denali, Eisai, Merry Life, Nervgen, Novo Nordisk, Optoceutics,

Passage Bio, Pinteon Therapeutics, Prothena, Red Abbey Labs, reMYND, Roche, Samumed, Siemens Healthineers, Triplet Therapeutics, and Wave, has given lectures in symposia sponsored by Alzecure, Biogen, Cellectricon, Fujirebio, Lilly, Novo Nordisk, and Roche, and is a co-founder of Brain Biomarker Solutions in Gothenburg AB (BBS), which is a part of the GU Ventures Incubator Program (outside submitted work). KB has served as a consultant, at advisory boards, or at data monitoring committees for Abcam, Axon, BioArctic, Biogen, JOMDD/Shimadzu. Julius Clinical, Lilly, MagQu, Novartis, Ono Pharma, Pharmatrophix, Prothena, Roche Diagnostics, and Siemens Healthineers, and is a co-founder of Brain Biomarker Solutions in Gothenburg AB (BBS), which is a part of the GU Ventures Incubator Program, outside the work presented in this paper. The remaining authors declare no competing interests.

## Additional information

[1]Department of Psychiatry and Neurochemistry, Sahlgrenska Academy, University of Gothenburg, Mölndal, Sweden. [2]Department of Neuroscience, Physiology and Pharmacology, University College London, Gower Street, London, United Kingdom. [3]Department of Neuropsychiatry, Sahlgrenska University Hospital, Mölndal, Sweden. [4]Department of Neurodegenerative Disease, Queen Square Institute of Neurology, University College London, Queens Square, London, United Kingdom. [5]Dementia Research Centre, Queen Square Institute of Neurology, University College London, Queens Square, London, United Kingdom. [6]Clinical Neurochemistry Laboratory, Sahlgrenska University Hospital, Mölndal Hospital, House V, Mölndal, Sweden. [7]Paris Brain Institute, ICM, Pitié-Salpêtrière Hospital, Sorbonne University, Paris, France. [8]Neurodegenerative Disorder Research Center, Division of Life Sciences and Medicine, and Department of Neurology, Institute on Aging and Brain Disorders, University of Science and Technology of China and First Affiliated Hospital of USTC, Hefei, P. R. China. [9]UK Dementia Research Institute at UCL, London, UK. [10]Hong Kong Center for Neurodegenerative Diseases, Clear Water Bay, Hong Kong, PR China. [11]Wisconsin Alzheimer's Disease Research Center, University of Wisconsin School of Medicine and Public Health, University of Wisconsin-Madison, Madison, WI, USA. [12]Department of Neurology, Feinberg School of Medicine, Northwestern University, Chicago, IL, USA. [13]These authors contributed equally: Jack I. Wood, Maciej Dulewicz. ✉e-mail: jh@gu.se

