## [Transparent Peer Review file · Nature Communications]

Isotope Encoded Spatial Biology Identifies Plaque-Age-Dependent Maturation and Synaptic Loss in an Alzheimer's Disease Mouse Model

Corresponding Author: Dr Jörg Hanrieder

Version 0:

Reviewer comments:

Reviewer #1

(Remarks to the Author)

Understanding the evolution of A β in Alzheimer's etiology is crucial for elucidating the mechanisms underlying disease progression, and this study partially addresses this aspect. The authors applied the iSILK technique to track amyloid plaque formation, maturation, neurotoxicity, and synaptic loss in a spatiotemporal way. The authors employ the AppNL-F/NL-F aged mouse model to examine how plaque formation progresses over time. The study presents an innovative approach by integrating mass spectrometry imaging (MALDI MSI), spatial transcriptomics, and hyperspectral microscopy to assess plaque heterogeneity and its associated effects. However, critical technical and conceptual issues need to be addressed before this manuscript can be considered for publication.

1. One major limitation of this study is the lack of consideration for known post-translational modifications (PTMs) of A β , which are known to influence its aggregation and pathological progression. The centroid mass values of MALDI MSI (Table S1) do not reflect known A β PTMs, which should result in characteristic mass shifts. The authors should respond the following questions regarding the peak assignment of their MALDI MS spectra.

- 1) Why were these known PTMs not detected in the dataset?
- 2) Does the dataset contain any mass peaks that could suggest age-dependent PTM changes in A β ?

2. The authors should explain why no background peptide signals are observed in the spectrum. Additionally, they should present the MS/MS spectra to confirm the presence of amyloid beta and assess whether other peptides were detected.

3. The authors claim that older amyloid plaques exhibit increased neurotoxicity and synaptic loss, as demonstrated through iSILK labeling, transcriptomics, and fluorescence imaging. While the data suggest a correlation between plaque age and synaptic dysfunction, it is not yet clear how this study advances our understanding beyond prior work that has already reported synapse loss in the vicinity of amyloid plaques.

To fully support their conclusions, the authors should clarify what new biological insights this study provides compared to previous findings on plaque-induced toxicity.

Reviewer #2

(Remarks to the Author)

Dear Editor and authors,

I have reviewed the paper "Isotope Encoded Spatial Biology Identifies Amyloid Plaque-Age-Dependent Structural Maturation, Synaptic Loss, and Increased Toxicity" by Hanrieder and co-workers. The paper aims to understand the initiation and progression of A β aggregate accumulation and correlations with CNS cell responses by spatial transcriptomics in the proximity of early and late A β aggregates. Chemical time stamps were induced using pulse chase of food supplemented with ¹⁵N-isotope that label newly produced protein facilitating using a method called Imaging of stable isotope labelling kinetics (iSILK). As a complementary time stamp, the maturity of the A β -plaque structures was monitored by LCO fluorescence technology developed by researchers at Linköping University, Sweden.

Spatial transcriptomics together with iSILK and LCO/immunofluorescence structure correlations allowed an unprecedented

correlation of proximal cell response to the plaque maturity stage. The study concluded that mature A β plaques, in comparison with early plaques, regardless of chronological mouse age, were associated with changes suggesting synaptic toxicity responses. Early plaque formations showed increased immune response gene expression. This methodological approach puts within reach more information on differentiated cellular responses to plaque fibril structure at different plaque development stages and cell types (microglia, astrocytes, and diverse neuronal populations).

Overall the impression of the innovative application of different integrated techniques and the presentation of the paper is very positive.

There are some of points that should be modified and clarified in the revised version to improve the paper:

1. The choice of the APPNL-F knock-in mouse in this study was clever, because this mouse makes almost exclusively A β 1-42, allowing the authors to specifically iSILK-monitor A β 1-42 species using MALDI-ToF imaging. This feature should be more clearly stated in the description of the selection of the mouse model. Suggestively at Results under the header "iSILK delineates spatial and structural patterns of plaque formation and maturation" after the sentence in the first paragraph "This gradual increase in plaque pathology with age more closely resembles the human disease". Add something like this: "The choice of the APPNL-F knock-in mouse in this study provided another biochemical advantage. Because this mouse makes almost exclusively deposited A β 1-42, it allows us to specifically iSILK-monitor A β 1-42 species using MALDI-ToF imaging".

2. Under the header "Amyloid plaque maturation is characterized by continuous fibrilization with age" There is a description of the LCO and especially qFTAA and hFTAA. The reference here is good but insufficient. LCO reference 21 is ok for general purposes but then the following (see below) should be added in the description. While the references to the original publications on this method are present in the paper (later in the discussion), from reading this result presentation it appears that the description comes at face-value from the present study.

So suggested placement of references: "This is enabled by the difference in affinity of the two LCO probes, q-FTAA and h-FTAA, towards amyloid aggregates. Specifically, q-FTAA preferentially binds to mature and compact beta-pleated aggregates, while h-FTAA binds to less compact, yet still beta-pleated aggregates (reference: Now ref 34 in the list of references Nystrom, S. et al. Evidence for age-dependent in vivo conformational rearrangement within Abeta amyloid deposits). Due to their different emission profiles, the LCO probes can be spatially delineated using hyperspectral fluorescent microscopy. Here, the ratio of the LCO maxima (500 nm for q- FTAA / 580 nm for h-FTAA) is used to express preferential binding of either of the two LCO probes used, whereby an increase in 500 nm intensity is indicative of increased q-FTAA binding and therefore, increased structural maturity of the amyloid fibrils (Fig. 2B). (reference: Now ref 35 Rasmussen, J. et al. Amyloid polymorphisms constitute distinct clouds of conformational variants in different etiological subtypes of Alzheimer's disease. + Now ref 45 Parvin, F. et al. Divergent Age-Dependent Conformational Rearrangement within Abeta Amyloid Deposits in APP23, APPPS1, and App(NL-F) Mice.).

3. The spacial transcriptomics data should be clarified. It is not clear how the reference data (baseline normalized expression) were obtained for making the volcano-plots of overexpression and decreased expression in Fig. 3C and 3F? In the same context: How to interpret the GeoMx CSV files when there is no reference data set included. Was this from already published databases? If so, the used numbers from this reference data set should be included. They can be very useful for others looking for other genes and to derive the data associated with the volcano plot (discussed above). The contrast (samples vs reference) is not obvious from the CSV-files. It appears to be only numbers listed for 10 and 18 month APPNL-F mice.

Furthermore, the link given:

<https://hanriederlab.shinyapps.io/PlaqueAgeTranscriptomics/>

can be very useful if it more clearly described how it was obtained and what cutoff values are used for significance. Is the x-axis plaque age (in months?) based on iSILK or q-FTAA/h-FTAA fluorescence and what is Log Cnorm? A clear description of how to use the link should be included in the paper with more descriptions in the methods section.

4. The list of acknowledgements for funding is very long. But there is no acknowledgement or information for where the authors obtained the LCOs q-FTAA and h-FTAA which plays an important role in the paper. Since these molecules appear not to be commercially available it needs to be specified. If they were synthesized in their own lab this should be stated in the materials and methods.

Reviewer #3

(Remarks to the Author)

In this manuscript, Wood & Dulewicz et al. describe the combination of mass spectrometry imaging (MSI) and hyperspectral imaging to measure Ab42 plaque age and morphology in a mouse model of Alzheimer's disease. Using a pulse-chase strategy where mice are fed with a 15N-enriched diet, they were able to distinguish nascent and aged plaques. Hyperspectral imaging with oligothiophene chemistry further demonstrates changes in plaque morphology correlated to age. They further combine this approach with single-plaque GeoMx spatial transcriptomics to identify correlates with age inferred from their MSI approach.

Overall, the experiments are executed quite well, and the methods and data will be useful for the spatial and Alzheimer's biology community. In the future, I also see great value in combining these techniques with higher-resolution spatial transcriptomics or multiomics to understand the molecular underpinning of these processes. While I find their iSILK MSI technology to be technically impressive, especially when combined with these orthogonal measurements, I believe the richness of their data could have been used to better extract biologically novel conclusions. The manuscript would also benefit from more extended analyses, embedding with more specific knowledge in literature, and better bookkeeping of the collected data. If the authors could address or comment on these points, I believe the manuscript will be significantly improved and I would be supportive of publication.

Major comments

1. The use of MSI pulse chase is quite inspired. I agree that 15/14N ratio is a good proxy for plaque age. Here are some points the authors may want to consider:

a. It seems like a missed opportunity for developing a numerical index or 'clock' here for plaque age. This could be used to assign ages to plaques without MSI, for instance. If hyperspectral imaging or morphology could be used to do a regression analysis, I think that could be immensely useful. I understand that this may be challenging to establish rigorously, but could be a point of discussion for future work.

b. It would be interesting for the authors to look at how the MSI and hyperspectral imaging replicates within and across animals. Are these correlations sufficiently strong and hence, biologically robust across animals? A breakdown of Fig. 2e by animal for example could be helpful.

c. Could the authors briefly comment on the feasibility and cost of deploying this technology? What is the feasibility as well of MSI and LCO imaging on the same section?

2. I have some general reservations about the GeoMx analysis. Given the richness of the dataset, the analysis should give correspondingly rich and novel biological insights. Broadly, are there any unexpected findings here?

a. Based on the sequencing, the authors suggest dysregulation of synaptic, immune and metabolic genes proximal to plaques, in a manner correlating with age on adjacent sections. Rather than a change in regulatory patterns, is it possible that these changes are explained instead by differences in cell population composition? Much of this is also driven by FOV (or rather, AOI) size and how many cells are expected to be captured.

i. Neuronal mRNA is typically enriched in the soma, with a number of exceptions, and often, in an activity-dependent manner. Thus, depletion of neurons might be an equally fair explanation of apparent decrease in synaptic gene expression observed in the GO analysis. This is likewise for immune/metabolic genes – as recently observed with high-resolution spatial data in 5XFAD mice. The authors could consider using morphological (Nissl, silver, DAPI etc.) staining or further immunofluorescence to quantify the cell populations here. Further markers for DAA/DAMs might also be helpful.

ii. In a similar vein, are there any changes in mRNAs known to be translocated to synapses, versus those known to be soma-localized?

iii. Do the authors observe upregulation of cryptic genes associated with nonnative cell types in these plaques? For instance, cell type markers of unexpected immune cells.

iv. Can the authors further embed some of these findings with more recent literature on functional changes in astrocytes and microglia?

b. The statistics on correlations between the GeoMx and plaque age should consider multiple testing correction if possible (q-val or FDR perhaps). My understanding is that raw p-values are shown. Are these conclusions still robust?

c. While the viewer is helpful, for some example genes, could the authors directly show the scatterplots demonstrating the correlations between gene expression and plaque age in the main text? Regressions with the corresponding coefficients might also be helpful.

d. Could the authors also provide some commentary on the challenges of hyperspectral imaging prior to GeoMx on the same tissue slice? As they mention, serial sections preclude the analysis of nascent small plaques, which would certainly be biologically fascinating. Do the authors see this as a major technical limitation? To my knowledge, there are several groups combining multimodal imaging and GeoMx imaging and sequencing on the same sections as well, so I am interested in just hearing some perspective on this – though an experimental demonstration would be most impressive.

3. There are some ways the data could also be better organized and bookkept, for the purpose of transparency:

a. Can the authors enumerate the number of plaques and their age distribution captured by MSI within and across animals?

b. Figure S1 is insufficiently described to let me understand the structure of the data. A more thorough caption and explanation would be helpful, including a breakdown by source animal rather than just age.

c. Could the authors clarify what is shown in Fig. S2b? I am not sure how to interpret these values. It might be more useful to show PCAs instead.

a. Could we have a complete description of all GeoMx FOVs imaged, and their corresponding $^{15}/^{14}\text{N}$ ratios, morphologies and hyperspectral ratios, anatomical localization, number of unique genes and transcripts sequenced, etc? It would also be important to know the source animals for each FOV. This is an essential quality control metric for readers to make critical judgments about the data. This could for example be in the form of a table (for small numbers of FOVs), or a heatmap. It will give me a better idea of how powered the conclusions are.

Minor comments

1. It might be helpful for the authors to further comment on the applicability of their method in conjunction with higher-resolution spatial transcriptomics methods (e.g. Visium HD, MERFISH, Xenium etc.) or proteomics (CODEX, DVP etc.) in future work. Integrating iSILK with the growing landscape of these tools is certainly a point of interest for many spatial biologists.

2. Some general commentary of the limitations of their approach might be helpful.

3. Fig. 1a scale bar value is not provided. The imaging modality should also be stated.

4. As a matter of preference, it might be better for authors to use perceptually uniform colormaps instead of spectral (e.g. in Fig. 1f). It may be hard for some readers to see.

5. Legends in some panels (e.g. Fig. 1e, 2c) also are rather small and hard to see.

6. Gene and GO annotations for Fig. 3c-h are also hard to read.

7. Could the MSI for Fig. 3b be decomposed into $^{15}/^{14}\text{N}$ for clarity?

8. In "spanning across two consecutive 12mm sections" did the authors means 12 micron?

Version 1:

Reviewer comments:

Reviewer #1

(Remarks to the Author)

I have now reviewed the revised manuscript, "Isotope Encoded Spatial Biology Identifies Amyloid Plaque-Age-Dependent Structural Maturation, Synaptic Loss, and Increased Toxicity," along with the authors' point-by-point rebuttal to the concerns I raised in my initial review.

The authors have addressed all of my previous concerns by conducting significant new experiments to validate their findings and strengthen their claims. The new high-sensitivity LC-MS/MS analysis to investigate post-translational modifications and the inclusion of MS/MS spectra for sequence confirmation have fully resolved my initial questions regarding the data.

As a final point of scientific interest, I would like to offer one minor point for the author's consideration. In the MS/MS spectrum for A β 1-42 shown in Supplementary Figure 1F, it is interesting to note the dominance of the b-ion series, while the corresponding y-ion series is almost absent. A brief comment on this observation in the Results section or the caption of the figure could enhance the spectral interpretation of the fragmentation behavior.

Reviewer #2

(Remarks to the Author)

The authors have made appropriate clarifications of all my previously raised points in the first round of detailed review and the paper has been improved. Congratulations to very interesting and important work. I support publication as soon as possible.

Reviewer #3

(Remarks to the Author)

The presentation of the data in the revised manuscript makes it much more straightforward to read and interpret the data in

my opinion. Here are some remaining minor comments for the authors, though they are mostly either presentation issues or nominal analyses that might provide some further insights if the authors choose to perform them. Nevertheless, I believe they have satisfied my major concerns, and I am supportive of publication.

Minor Comments

1. In their discussion of limitations of other high-res ST approaches, they might note that the CosMx 18k now offers near-transcriptome coverage – though at markedly lower gene-wise sensitivity
2. I would appreciate if the authors could do one additional analysis – to highlight novel gene-gene correlation modules that arise across from analysis across plaques (e.g. with WGCNA or similar).
3. The authors might comment on whether their observed gene expression changes signify some T cell infiltration (or any other evidence for this in their animals) versus microglial activation due to overlap in GO annotations. This was what I meant by cryptic expression in my previous review and I apologize for the misunderstanding.
4. The statement that APP NL-F is strictly better than NL-G-F as it reflects human pathology (Line 387-389) is too strongly made.
5. Scale bars missing in Fig. 3A, 4A.
6. Labels for GFAP, Ab, and SYTO colors are missing in Fig. 3B, Supp Fig. 5, 6.
7. I am not sure why the data must be duplicated as Supp Fig. 3D – would be preferable to instead include these panels in Fig. 3B?
8. The bottom axis labels for Fig. 3C, D are clipped.

Typographical

1. Line 190 “oligothiophene (LCO)” > “oligothiophenes (LCOs)
2. Line 429 “nformation” > “information”

Dear Reviewers

Please find attached with this resubmission our revised manuscript “Isotope Encoded Spatial Biology Identifies Amyloid Plaque-Age-Dependent Structural Maturation, Synaptic Loss, and Increased Toxicity” by et al. We are grateful for the overall positive, thorough and constructive feedback provided, which helped significantly to both clarify and improve the manuscript substantially.

Please find below a detailed point-by-point response to the comments raised by the reviewers. The corresponding changes have been highlighted in an additional manuscript file (article_tracked_changes).

With best regards
Jörg Hanrieder

Reviewer #1:

Comment 1: “One major limitation of this study is the lack of consideration for known post-translational modifications (PTMs) of A β , which are known to influence its aggregation and pathological progression. The centroid mass values of MALDI MSI (Table S1) do not reflect known A β PTMs, which should result in characteristic mass shifts. The authors should respond to the following questions regarding the peak assignment of their MALDI MS spectra.”

(1) Why were these known PTMs not detected in the dataset?

We concur that this is an important point. Unfortunately, it is a limitation of this mouse model that no prominent A β truncations or PTMs are observed, specifically pE-x and PSer8-x (Saito et al. 2014). To underpin this observation, we performed additional, in-depth ex situ mass spectrometric experiments to investigate potential sequence truncations/modifications that might be suppressed or below the detection limit in the MALDI MSI analyses.

Specifically, we performed two elaborate ex situ analyses of brain tissue extracts, whole proteomic profiling of purified fibrils and immune-precipitation LC-MS/MS of amyloid peptides.

To this end, we first immune-precipitated A β peptides from App^{NL-F} brain extracts and analyzed the samples by LC-MS/MS on a high-resolution Orbitrap mass spectrometer (Supplementary Fig. 1). Resulting data was searched in PEAKS (de novo peptide sequencing) for characterizing potential posttranslational modifications, including phosphorylation, deamidation, oxidation and pyroglutamation. The analysis confirmed the presence of A β 1-38, 1-39, and 1-42, with 1-42 being the most prominent species. Following sequence analyses, no PTMs were observed for either species. Please find an excerpt from the obtained PEAKS results, now detailed in Supplementary Fig. 1B, in the table below:

Table 1: Excerpt from PEAKS results of the anti-A β IP-LC-MS/MS experiment.

Gene name	Peptide	Mass	Length	ppm	m/z	z	RT	PTM
APP	DAEFRHDSGYEVHHQKLVFFAEDVGS NKGAIIGLMVGG	4129,012	1-38	3,1	826,81213	5	43,179	/
APP	DAEFRHDSGYEVHHQKLVFFAEDVGS NKGAIIGLMVGGV	4228,08	1-39	2,5	846,62537	5	44,834	/
APP	DAEFRHDSGYEVHHQKLVFFAEDVGS NKGAIIGLMVGGVIA	4511,27	1-42	1,1	903,26215	5	68,445	/

The obtained RAW data and results have been uploaded to the PRIDE repository under the accession number PXD063759.

Second, we analyzed fibrils purified from NLF brain. For this we used proteomics as the enzymatic digestion step (trypsin) provides increased sensitivity towards potentially truncated species. In contrast, this approach does not necessarily provide exact annotation of N-terminal modification for different C-terminal species (i.e x-40, x-42). Similarly to the IP-MS analyses, no PTMs were observed for the A β fragments. The corresponding raw data is accessible on the PRIDE repository under the accession number MSV000092311.

All experiments described above are now detailed in methods section “Ex situ A β Analysis” and have been incorporated into the main text on page 25 line 24.

(2) Does the dataset contain any mass peaks that could suggest age-dependent PTM changes in A β ?”

We thank the reviewer for this relevant question. Neither our MALDI MSI nor LC-MS/MS experiments (see Table above and Supplementary Fig. 1) suggest any presence of PTM for A β .

In our MALDI MS data, the phosphorylation would be detectable as the mass shift is +98 (H₂PO₂) or +80 (PO₃), while a shift in 15N can be maximal +55 (as there are only 55 nitrogen in the A β 1-42 sequence). Similarly, the 3pE-x or 11pE-x modification would require any peak close to the unmodified 3-x and 11-x, neither of which is observed.

For the raw LC-MS/MS data, please refer to the PRIDE repository (accession numbers PXD063759 and MSV000092311), and for the raw MALDI MSI data, see the Shiny app: (https://maciejdulewiczgu.shinyapps.io/MALDI_GEOMX_VOLCANO/), where we present the data in an easily accessible format for the public.

Please see Supplementary Fig. 1.

Comment 2: “The authors should explain why no background peptide signals are observed in the spectrum.

The focus of the current work was on imaging amyloid peptides that are inherently difficult to detect in situ due to their hydrophobic properties and highly aggregated state. Consequently, we have been tuning our protocols to enhance the amyloid signal, which requires extensive washes and formic acid hydrolysis. This extensive sample

preparation leads, in turn, to a reduced signal of other less aggregated, endogenous peptides and small proteins. We performed control experiments with and without formic acid to demonstrate the effects of FA retrieval on amyloid signal, as shown below in Figure 1.

This is now clarified in the methods on page 22 line 20 and in a newly created Supplementary Fig. 2.

“Additionally, they should present the MS/MS spectra to confirm the presence of amyloid beta and assess whether other peptides were detected.”

We thank the reviewer for this comment and agree that MS/MS confirmation of A β presence is important. Due to the limited sensitivity of direct in situ MALDI MS/MS, we employed extensive ex situ analyses for A β sequence validation as described above under Comment 1. Briefly, we performed anti-A β immunoprecipitation followed by LC-MS/MS as well as LC-MS/MS-based proteomics of purified amyloid fibrils, all on App^{NL-F} mouse brain tissue extracts. Here, LC-MS/MS confirmed the presence of A β 1-42, along with A β 1-38 and A β 1-39. A representative MS/MS spectrum of A β 1-42 is shown in Supplementary Figure 1F.

Our MALDI MSI data predominantly show A β 1-42 within plaques, with A β 1-38 detected in only two ROIs/plaques (low S/N ratio), while 1-39 was not detected in any MALDI MSI ROI. This further highlights that A β 1-42 is indeed the predominant A β species. The identification of A β 1-38 and A β 1-39 by LC-MS/MS is not unexpected and rather highlights the high sensitivity of this method (as compared to direct in situ detection via MALDI MSI).

Please see Supplementary Figure 1.

Comment 3: “The authors claim that older amyloid plaques exhibit increased neurotoxicity and synaptic loss, as demonstrated through iSILK labeling, transcriptomics, and fluorescence imaging. While the data suggest a correlation between plaque age and synaptic dysfunction, it is not yet clear how this study advances our understanding beyond prior work that has already reported synapse loss in the vicinity of amyloid plaques. To fully support their conclusions, the authors should clarify what new biological insights this study provides compared to previous findings on plaque-induced toxicity.”

We appreciate this comment and would like to offer an extended explanation and discussion. Most importantly, the methods described allow us to decouple plaque age from chronological age of the mice. This brings several advantages that would not be discernible with static methods.

1) Most importantly, this reveals and validates that plaques indeed affect synapses and that persisting plaque ageing/maturation is associated with proximal synaptic dysfunction and loss, rather than plaques exerting immediate, acute toxicity. In static experiments one could not discern the age of individual plaques within one mouse/age group. Hence it would not be possible to assess whether the differences in plaque phenotypes and their effects (for example on synapses) are a result of plaque maturation or whether they represent different subgroups of plaques within a single tissue. Our methods clearly allow for this distinction.

2) Even more important, on the transcriptomic scale, the effect of plaque maturation is easily masked when comparing plaques with each other across the same mouse ages. For this, we further demonstrate the advantage of correlation of similar, spatially refined entities as compared to both bulk RNAseq and static spatial transcriptomics (ST) experiments. This is demonstrated in the more sensitive detection of synaptic genes that vary across plaques within the same mouse. To highlight this, we provide additional data demonstrating how both spatial transcriptomics (as compared to bulk brain extract transcriptomics), as well as dynamic spatial transcriptomics, improve sensitivity and specificity to plaque and plaque age molecular processes (please see Supplementary Fig. 10).

3) We further demonstrate conceptual novelty in integrating chemical imaging with spatial transcriptomics. Of note, this is demonstrated by the combination of functional LCO microscopy with iSILK and ST, showing that plaque maturation involves the continuous fibrillization at the plaque.

4) An additional advance is related to the technical innovation, implementing the iSILK method in old mice. While challenging, we succeeded in following plaque formation from before plaques form (6-10mo) during plaque growth at old ages (18mo). This showed that for this knock in model, plaques form as small cores in the cortex consisting of 1-42.

To clarify the novelty of our biological insights, we have added the following sentences to the discussion and embedded references to the most recent and impactful spatial transcriptomics studies in the field (page 15 line 18):

“Whereas findings from other impactful spatial transcriptomics studies on AD brain tissue (Chen et al. 2020, Mallach et al. 2024) have also supported accumulated rather than acute plaque toxicity, they were unable to directly link plaque age with increasing synaptic dysfunction, entirely decoupled from chronological mouse age. In fact, many studies have focused on the spatial rather than temporal aspect of plaque toxicity (Wood et al. 2022, Johnston et al. 2025) as well as focused on subsets of genes associated with e.g. glial activation in response to plaque pathology. Our study, in contrast, allowed us to evaluate the effects of maturing plaques independent of mouse age on a whole transcriptomic scale, with key findings being validated with direct immunological stainings. Additionally, the mouse model used in this study more accurately reflects human AD as this combines WT Abeta pathology together with ageing, which contrasts with the NLGF mouse model, on which most prior conclusions have been based.”

Reviewer #2:

Dear Editor and authors, I have reviewed the paper “Isotope Encoded Spatial Biology Identifies Amyloid Plaque-Age-Dependent Structural Maturation, Synaptic Loss, and Increased Toxicity” by Hanrieder and co-workers. The paper aims to understand the initiation and progression of A β aggregate accumulation and correlations with CNS cell responses by spatial transcriptomics in the proximity of **early** and late A β aggregates. Chemical time stamps were induced using pulse chase of food supplemented with ¹⁵N-isotope that label newly produced protein facilitating using a method called Imaging of stable isotope labelling kinetics (iSILK). As a complementary time stamp, the maturity of the A β -plaque structures was monitored by LCO fluorescence technology developed by researchers at Linkoping University, Sweden. Spatial transcriptomics together with iSILK and LCO/immunofluorescence structure correlations allowed an unprecedented correlation of proximal cell response to the plaque maturity stage. The study concluded that mature A β plaques, in comparison with early plaques, regardless of chronological mouse age, were associated with changes suggesting synaptic toxicity responses. Early plaque formations showed increased immune response gene expression. This methodological approach puts within reach more information on differentiated cellular responses to plaque fibril structure at different plaque development stages and cell types (microglia, astrocytes, and diverse neuronal populations). Overall the impression of the innovative application of different integrated techniques and the presentation of the paper is very positive. There are some of points that should be modified and clarified in the revised version to improve the paper:

Comment 1: “The choice of the APP NL-F knock-in mouse in this study was clever, because this mouse makes almost exclusively A β 1-42, allowing the authors to specifically iSILK-monitor A β 1-42 species using MALDI-ToF imaging. This feature should be more clearly stated in the description of the selection of the mouse model. Suggestively, at **Results** under the header ‘iSILK delineates spatial and structural patterns of plaque formation and maturation’ – after the sentence in the first paragraph ‘This gradual increase in plaque pathology with age more closely resembles the human disease’ – add something like: “The choice of the APP NL-F knock-in mouse in this study provided another biochemical advantage. Because this mouse makes almost exclusively deposited A β 1-42, it allows us to specifically iSILK-monitor A β 1-42 species using MALDI-ToF imaging.”

We are grateful for these positive comments and the constructive feedback. We concur with this comment regarding the choice of the mouse model. As suggested, a corresponding statement, commenting on the biochemical advantage of the mouse model, has been incorporated in the results. In addition, we have performed additional ex situ LC-MS/MS experiments confirming A β 1-42 to be the predominantly produced and deposited A β species. Please see Supplementary Fig. 1, Methods section “Ex situ A β Analysis” and Results page 25 line 24.

Comment 2: *“(Under the header ‘Amyloid plaque maturation is characterized by continuous fibrillization with age’ there is a description of the LCO probes q-FTAA and h-FTAA. The reference here is good but insufficient. LCO reference 21 is ok for general purposes but the following should be added in the description (with appropriate references):

'This is enabled by the difference in affinity of the two LCO probes, q-FTAA and h-FTAA, towards amyloid aggregates. Specifically, q-FTAA preferentially binds to mature and compact β -pleated aggregates, while h-FTAA binds to less compact, yet still β -pleated aggregates (reference: now ref 34, Nyström et al. Evidence for age-dependent in vivo conformational rearrangement within A β amyloid deposits). Due to their different emission profiles, the LCO probes can be spatially delineated using hyperspectral fluorescent microscopy. Here, the ratio of the LCO maxima (500 nm for q-FTAA / 580 nm for h-FTAA) is used to express preferential binding of either of the two LCO probes, whereby an increase in 500 nm intensity is indicative of increased q-FTAA binding and therefore increased structural maturity of the amyloid fibrils (Fig. 2B).'

And then add references for these statements, e.g., ref 35 Rasmussen et al. (Amyloid polymorphisms...), and ref 45 Parvin et al. (Divergent Age-Dependent Conformational Rearrangement...)."

We appreciate these suggestions and revised this part accordingly (page 8, lines 18).

Comment 3: *"The spatial transcriptomics data should be clarified. It is not clear how the reference data (baseline normalized expression) were obtained for making the volcano plots of overexpression and decreased expression in Fig. 3C and 3F.*

*In the same context: How does one interpret the GeoMx CSV files when there is no reference data set included? Was this from an already published database? If so, the values from this reference data set should be included, as they can be very useful for others looking at other genes and to derive the data underlying the volcano plot. The contrast (samples vs reference) is not obvious from the CSV files – it appears to be only numbers listed for 10-month and 18-month *App*^{NL-F} mice.*

We agree with this important comment and would like to further explain what is shown in the data. The volcano plots 3C and 3F do not represent differential expression analysis. They show genes that were significantly correlating with plaque age, both positively and negatively for the respective mouse age group (10mo: 3C; 18mo: 3D). Consequently, there is no reference data set to refer to.

We acknowledge that this is not clearly presented in the manuscript. We therefore extended the description of the data in Figure legend 3.

Additionally, we adjusted the wording in the main text on page 10 line 14 for clarification: "Volcano plots for both 10-month (Fig. 3C) and 18-month-old (Fig. 3D) mice demonstrated that gene expression levels had significant positive and negative correlations with increasing plaque age."

Furthermore, the link given (the Shiny app: <https://hanriederlab.shinyapps.io/PlaqueAgeTranscriptomics/>) can be very useful if it is more clearly described how it was obtained and what cutoff values are used for significance.

We agree and provide a detailed description of the data accessible in the Shiny App both in the methods and additionally detailed in a Readme file to highlight the details of the data and the analysis they originate from. Additionally, we updated the methods section "MALDI MSI - GeoMx Data Analysis" for increased clarity, including cut-off values. The data accessible via the ShinyApp is briefly explained under the Data

*Availability section in the manuscript. Also, we curated all GeoMx and MALDI data under a new ShinyApp address. Please see:
https://maciejdulewiczgu.shinyapps.io/MALDI_GEOMX_VOLCANO/*

Is the x-axis “plaque age” (in months?) based on iSILK or q-FTAA/h-FTAA fluorescence, and what is “**Log Cnorm**”? A clear description of how to use the link should be included in the paper, with more details in the methods section.”*

We concur that this is critical and update the description of these annotations throughout all presented Figures. In brief, the x-axis values are representing measures of plaque age and are obtained from calculating the width of the MALDI mass peak for Ab 1-42. Here the width (calculated as full width at half maximum, FWHM) is a measure of peak broadening and consequently an indication of ¹⁵N incorporation and plaque age, respectively.

In Figures 1-4 as well as in the Shiny app, the x-axis is now referred to as “Nitrogen index LP”, which is explained in the main text on page 10 lines 9 and page 30 line2 and visualized in the updated Fig. 1 E.

We also included a more detailed explanation of the nitrogen index calculation under the header “Calculation of nitrogen index in linear positive mode” in the methods section.

Comment 4: “The list of acknowledgements for funding is very long. But there is no acknowledgement or information for where the authors obtained the LCOs q-FTAA and h-FTAA, which play an important role in the paper. Since these molecules do not appear to be commercially available, it needs to be specified. If they were synthesized in their own lab this should be stated in the Materials and Methods.”

We received the probes as a kind gift. This information was updated in the methods and acknowledgements.

Reviewer #3:

In this manuscript, Wood & Dulewicz et al. describe the combination of mass spectrometry imaging (MSI) and hyperspectral imaging to measure Ab42 plaque age and morphology in a mouse model of Alzheimer's disease. Using a pulse-chase strategy where mice are fed with a ^{15}N -enriched diet, they were able to distinguish nascent and aged plaques. Hyperspectral imaging with oligothiophene chemistry further demonstrates changes in plaque morphology correlated to age. They further combine this approach with single-plaque GeoMx spatial transcriptomics to identify correlates with age inferred from their MSI approach.

Overall, the experiments are executed quite well, and the methods and data will be useful for the spatial and Alzheimer's biology community. In the future, I also see great value in combining these techniques with higher-resolution spatial transcriptomics or multiomics to understand the molecular underpinning of these processes. While I find their iSILK MSI technology to be technically impressive, especially when combined with these orthogonal measurements, I believe the richness of their data could have been used to better extract biologically novel conclusions. The manuscript would also benefit from more extended analyses, embedding with more specific knowledge in literature, and better bookkeeping of the collected data.

If the authors could address or comment on these points, I believe the manuscript will be significantly improved and I would be supportive of publication.

We are grateful for the positive feedback and appreciate the constructive comments that helped to significantly improve the manuscript. Please see our detailed responses below.

Major Comments:

Comment 1: *"The use of MSI pulse-chase is quite inspired. I agree that the $^{15}\text{N}/^{14}\text{N}$ ratio is a good proxy for plaque age. Here are some points the authors may want to consider:*

- a. *It seems like a missed opportunity for developing a numerical index or 'clock' for plaque age. This could be used to assign ages to plaques without MSI, for instance. If hyperspectral imaging or morphology could be used to do a regression analysis, I think that could be immensely useful. I understand that this may be challenging to establish rigorously, but it could be a point of discussion for future work."*

This is a very important point, and we would like to clarify some methodological aspects as we concur and were in fact aiming to use the LCO as surrogate markers of plaque age. In detail, we performed two correlative spatial experiments (Fig. 1D-E):

1) iSILK (MALDI MSI) in conjunction with spatial transcriptomics (Fig. 1E) to investigate plaque age-specific changes in gene expression.

2) iSILK (MALDI MSI) in combination with LCO hyperspectral microscopy (Fig. 1D; though the LCO microscopy was not performed together with GeoMx for limited feasibility reasons, we further detail below under Comment 2d.)

The rationale for combining iSILK (MALDI MSI) with LCO microscopy was to both identify whether aged plaques are characterized with changes in fibrilization and,

exactly as suggested by your comment, to have a surrogate marker of plaque age that is easier to implement with follow-up experiments and does not require costly and time-consuming SILK labelling experiments. (Indeed, the LCO approach has been used for the IHC experiments described in Fig. 4, experimental overview in Fig. 1F)

Please refer to an updated Figure 1 for a detailed methodological overview.

As suggested by the reviewer, we incorporated this in the discussion on page 19 line 4, highlighting the future potential of the LCO q/h emission ratio.

b. “It would be interesting for the authors to look at how the MSI and hyperspectral imaging replicate within and across animals. Are these correlations sufficiently strong and hence biologically robust across animals? A breakdown of Fig. 2E by animal, for example, could be helpful.”

We thank the reviewer for raising this important question. In Fig. 2E, the correlation between MSI and LCO hyperspectral imaging was originally presented for each individual mouse (n = 3, with each plot corresponding to one mouse). This showed that the correlation is biologically robust across animals included in the study (R = 0.64-0.89, p<0.05)

We acknowledge that this was not clearly described and improperly visualized in the original submission.

To improve clarity, we have now combined all plaques from the three mice into a single plot (Fig. 2E) and revised the figure caption accordingly. Individual mice are distinguished by color, and plaque location (hippocampal vs. cortical) as indicated by different shapes. Both the combined trend (Fig. 2F) and the individual mouse correlations (now reported in the main text on page 9, line 10, Supplementary Fig. 4A-C) are consistent and support the same conclusions:

a) the hyperspectral ratio, reflects plaque age, b) the plaque core is older than the plaque periphery and c) cortical plaques are, on average, older than hippocampal plaques.

c. “Could the authors briefly comment on the feasibility and cost of deploying this technology? What is the feasibility as well of MSI and LCO imaging on the same section?”

We appreciate this comment as there are certain challenges and limitations that come with the various spatial techniques that need to be considered when planning the experiments.

A significant cost factor for iSILK is the stable isotope diet (ca \$12000/kg). This is particular true for the long labelling experiments needed in the present study as those mice develop amyloid pathology gradually with age.

Each mouse requires 1g labelled diet per day. We labelled each mouse for 4months (ca 122days ie 122g/mice, n=7) requiring 1kg label for all 7 mice. In addition, there are of course breeding costs and most importantly costs for personell as these experiments pan over 1-2years.

In addition, buffers and probe reagents constitute a major cost for the associated GeoMx experiments. GeoMx kits and buffers for two glasses amount to ca. \$3000/glass slide (3-4 sections/glass). Finally, sequencing costs to quantify the released probes amount to ca. \$3000/index plate. (96 AOI/plate). As those latter costs are highly dependable on inhouse availability and deals, we merely mentioned the

consumables (diet and glasses/GeoMx run in the Methods section/Experimental Design and Spatial Transcriptomics).

A further challenge includes further the compatibility of the different techniques for correlative spatial biology/multiomics in a single section. The LCO microscopy requires mounting with cover slips which prevents to do LCO imaging prior to MALDI (or GeoMx). In turn, the MALDI sample preparation and acquisition settings for peptide imaging lead to tissue distortions that prevent subsequent fluorescent microscopy. (please see Kaya et al 2017 PMID: 28318232).

Consequently, the MALDI/LCO as well as the MALDI/GeoMx analyses had to be carried out on sequential sections.

We further discuss compatibility challenges in the limitations (page 19 line 8) and provide more detail on the design of our correlative experiments in a revised Fig 1.

Comment 2: “I have some general reservations about the GeoMx analysis. Given the richness of the dataset, the analysis should give correspondingly rich and novel biological insights. Broadly, are there any unexpected findings here?”

We thank the reviewer for raising this concern regarding the richness of the results.

While the dataset is indeed rich, capturing a total of 19962 RNA transcripts, the lack of even more distinct and rich changes can be related to the design of the study and the research question we wanted to address i.e. Is the toxicity of plaques an immediate acute phenomenon in the area that they occupy or do plaques continue to cause ongoing increasing toxicity as they mature over time.

Previous studies on plaque associated changes in gene expression did not consider the time component on a single plaque age level and in addition rather focus on differential changes between plaques and non-plaque regions. Indeed, as Supplementary Figure 10 demonstrates, a direct spatial transcriptomics comparison between plaque-associated and non-plaque-associated areas revealed distinct up- or downregulation of disease-associated astrocytic and synaptic genes. These marked changes likely reflect the stark biological contrast between two fundamentally different environments: one area containing neurotoxic plaques and one area without plaque pathology.

Our study, in contrast, aimed to explore how gene expression correlates with increasing plaque age, independently of the animals' chronological age (something warranted by our iSILK paradigm). Given the relative similarity between differentially aged though biochemically similar cored plaques our plaque centric GeoMx analyses was naturally expected to reveal rather subtle biological differences as compared to analyses where plaque changes are compared to non-plaque regions.

It was very interesting and surprising that the major groupings affected in this incremental way, correlating with plaque age, were the synaptic genes.

To our knowledge, we are the first to demonstrate this continuous decline in synaptic gene expression with increasing plaque age on a (whole) transcriptomic scale. Thus, we can show that the well-known synaptic loss at plaques is not an immediate acute

effect but rather a slow ongoing infliction of stress and damage as the plaque matures over time.

This is now more clearly presented and compared to relevant literature in the discussion on page 15 line 18.

a. Based on the sequencing, the authors suggest dysregulation of synaptic, immune, and metabolic genes proximal to plaques, correlating with plaque age on adjacent sections. Rather than a change in regulatory patterns, is it possible that these changes are explained instead by differences in cell population composition? Much of this is also driven by FOV (AOI) size and how many cells are captured.

- i. Neuronal mRNA is typically enriched in the soma (with some exceptions, often in an activity-dependent manner). Thus, depletion of neurons might be an equally fair explanation of the apparent decrease in synaptic gene expression observed in the GO analysis. This likewise applies for immune/metabolic genes – as recently observed with high-resolution spatial data in 5xFAD mice. The authors could consider using morphological (Nissl, silver, DAPI, etc.) staining or further immunofluorescence to quantify the cell populations here. Further markers for DAA/DAMs might also be helpful.*

We concur and would like to offer our perspective along with additional data analyses addressing this concern, which is that differential cellular composition of the AOI can yield artefacts rather than true biological changes.

Delineating cell population heterogeneity is in fact a limitation of GeoMx as cell-type-specific analysis at the single-plaque level is currently not practically feasible using this method. The number of transcripts from individual cell types surrounding a single plaque within a finite AOI is insufficient to support high-quality transcriptomic profiling. Alternatively, pooling cell types from multiple plaques would result in a loss of spatiotemporal resolution, including the ability to associate transcriptomic data with individual plaque age. Additionally, GeoMx experiments are limited by the number of morphological markers that can be used simultaneously (typically 3 to 4), which restricts cell-type identification, particularly when pathological structures like A β plaques, as in our study, are also being imaged. Consequently, we were limited by the morphology markers we managed to implement (A β , cell count (SYTO, similar to DAPI) and GFAP).

With this we were however able to evaluate any plaque age associated changes in AOI cell count (SYTO), AOI size and GFAP immunoreactivity.

First, there was no significant correlation ($P > 0.05$) between AOI size or SYTO cell count with plaque age (Supplementary Fig. 8), suggesting that transcriptional differences are unlikely to be a result of systematic differences in the number of cells surrounding aging plaques.

Second, to estimate the number of astrocytes near plaques within the AOIs, we quantified the GFAP signal area for each AOI. Notably, correlation analysis between

Gfap mRNA expression and GFAP IHC signal showed strong concordance ($R=0.95-0.97$, $P<0.0001$; Supplementary Fig. 9 A-B), indicating that Gfap transcript levels indeed reflect corresponding protein abundance.

To investigate whether astrocyte numbers change with increasing plaque age, we correlated the GFAP IHC signal with plaque age. However, no significant correlation was observed ($P>0.05$, Supplementary Fig. 9 C-D), suggesting that astrocyte numbers do not systematically vary with plaque age. However, since GFAP is generally regarded as a marker of proliferation rather than activation, this lack of correlation does not necessarily imply an absence of astrocytic reactivity to plaques. It is possible that astrogliosis (i.e., astroglial proliferation) is more prominent during initial A β plaque formation and may diminish as plaques age. It is also important to stress that we might be underpowered to detect meaningful differences in astrogliosis with increasing plaque age.

A discussion of this limitation to not capture potential cell-type heterogeneity in the proximity of aging plaques as well as the Gfap IHC has been added to the discussion (page 17 line 7) and in the limitation (page 19 line 8).

ii. In a similar vein, are there any changes in mRNAs known to be translocated to synapses, versus those known to be soma-localized?

This is a very interesting point as it would be interesting whether the lower number of synaptic genes with plaque age is a consequence of synapse specific impairment or global neuronal loss around plaques.

We therefore analyzed mRNAs that were significantly positively or negatively correlated with plaque age to assess their subcellular localization (soma vs. neuropil, i.e., synaptic regions), For this we used data from Glock et al. (2021)(PMID: 34670838) that identified 807 neuropil-translocated (28%) and 2945 (72%) soma-located RNAs (3.5-times more soma- than neuropil-localized transcripts).

Our data show a similar pattern: across both positively and negatively correlated gene sets, we found 4–5 times more soma- located than neuropil-translocated RNAs (please see Table below).

mouse age (directionality of correlation)	Localization	Number of Genes (n)	Percentage of all significantly correlated RNAs in respective group (%)
10 m (positive)	neuropil	7	20
	soma	28	80
10m (negative)	neuropil	35	16,6
	soma	176	83,4
18 m (positive)	neuropil	24	17,8
	soma	111	82,2

18 m (negative)	neuropil	55	25,3
	soma	162	74,7

An exception to this trend was observed in genes negatively correlated with plaque age in 18-month-old mice, where 25% were neuropil-translocated (2.9-fold more soma- than neuropil-localized RNAs).

In comparison, only 16% of negatively correlated genes were neuropil-translocated in 10-month-old mice (a 5-fold difference).

This could suggest that with increasing plaque age, synaptic loss becomes more pronounced relative to general neuronal loss.

However, we emphasize that this interpretation is speculative due to several limitations: (1) the classification of RNA localization is based on a single study, as transcriptome-wide RNA translocation is not well-characterized; (2) we cannot rule out (post-mortem) RNA diffusion or other artifacts affecting localization; and (3) the data do not allow for robust statistical comparison.

We included a comment on this in the discussion. (page 16 line 12)

iii. Do the authors observe upregulation of **cryptic genes** associated with non-native cell types in these plaques? For instance, cell type markers of unexpected immune cells.

We thank the reviewer for this interesting question. According to Bruker Spatial Biology, GeoMx probes are designed not to target cryptic genes, as they are based on the RefSeq transcriptome, which includes only well-characterized transcripts.

iv. Can the authors further embed some of these findings with more recent literature on **functional changes in astrocytes and microglia?**

We agree and in order to address this, present correlation data for disease associated microglial and astroglial genes in both Fig 3 and Supplementary Fig. 7 in the Results and discuss those data with respect to the current literature (page 17 line 7).

Comment 2b: “The statistics on correlations between the GeoMx data and plaque age should consider multiple testing correction if possible (q-value or FDR perhaps). My understanding is that raw p-values are shown. Are these conclusions still robust?”

*The reviewer is correct: the volcano plots in Figures 3C and 3F display unadjusted p-values. Correlations were performed across a total of 34 plaques, which limits statistical power. Consequently, the resulting p-values do not survive transcriptome-wide FDR correction. A key limitation of applying correction, such as the commonly used Benjamini-Hochberg method, for multiple comparisons in genome-wide **correlation analyses** rather than the common simple pairwise comparisons (across 19,963 genes) is that the threshold for statistical significance becomes extremely stringent. Specifically, only near-perfect correlations ($r \approx 0.9$) will survive correction.*

We fully acknowledge the reviewer's concern regarding robustness. However, we believe our broader findings (namely, the decrease in synaptic gene expression and the increase in inflammatory and metabolic gene expression with increasing plaque age) remain robust.

This conclusion is supported by the consistent correlation patterns observed not in isolated genes, but across entire groups of synaptic, inflammatory, and metabolism-related genes, suggesting biologically meaningful and coherent changes.

Importantly, all GO analyses presented in the manuscript were performed using FDR-corrected p-values.

We have now explicitly noted this limitation (limited statistical power) in the discussion section (page 19, line 8) and specified in the methods section (page 33 line 6) and Figure 3 caption that uncorrected p-values are displayed for the correlation analyses.

Comment 2c: “While the viewer is helpful, **for some example genes**, could the authors **directly show the scatterplots demonstrating the correlations between gene expression and plaque age in the main text?** Regressions with the corresponding coefficients might also be helpful.”

We concur and include representative scatterplots for different synaptic (Nptx2, Dlg4) and DAA/DAM genes (Anxa1, Axl, Csf1, Ctsd) in Figure 3 F,G,I,J,L,M,O,P and Supplementary Fig. 7.

Comment 2d: “Could the authors provide some commentary on the challenges of hyperspectral imaging prior to GeoMx on the same tissue slice? As they mention, serial sections preclude the analysis of nascent small plaques, which would certainly be biologically fascinating. Do the authors see this as a major technical limitation? There are groups combining multimodal imaging and GeoMx on the same sections as well, so I am interested in hearing some perspective on this – though an experimental demonstration would be most impressive.”

As rightfully acknowledged by the reviewer, the LCO microscopy provides the possibility to serve as surrogate marker for plaque maturation/age and provides additional biophysical insight on plaque constitution. Consequently, it would be desirable to perform LCO and GeoMx within the same tissue rather than or in addition to, using a correlative approach using sequential sections. The same applies for combining MALDI and GeoMx.

However, LCO-based hyperspectral microscopy requires the use of mounting media and cover slips to afford an accurate readout of the spectral data and linear unmixing, respectively. Therefore, these experiments cannot be performed prior to the GeoMx (or MALDI) experiments on the same tissue section. (please see our response to your comment 1c)

We naturally investigated whether LCO staining could be performed after MALDI or GeoMx as well as whether MALDI and GeoMx could be performed on the same tissue. For MALDI, the combination with LCO microscopy and/or GeoMx is not possible as the sample prep and MSI peptide experiments leads to significant tissue distortion. (Of note, this is specific for amyloid peptide MSI. MALDI or DESI MSI of lipids could theoretically be interfaced with direct LCO imaging or GeoMx, as those MSI methods show no tissue distortion, please see Kaya et al 2017 doi:10.1021/acs.analchem.7b00313/ PMID: 28318232)

Regarding LCO analysis after GeoMx: this approach while feasible has as obvious limitations with respect to AOI selection as the LCO data should preferably guide AOI selection prior to the GeoMx experiment and not retrospectively.

Further, as mentioned under Comment 1a, the GeoMx approach is limited with respect to emission wavelengths used for fluorescent imaging as the UV range (<420nm) cannot be used due to photocleavable tags used that absorb in that range.

In addition, the LCO dyes emit rather broadly between 450-620nm, which leads to interference with morphology markers used in the GeoMx experiment that emit in this range.

These challenges are now discussed in the limitation section at the end of the discussion. (page 19 line 8)

Comment 3: “There are some ways the data could be better organized and bookkept, for transparency:

a. Can the authors enumerate the number of plaques and their age distribution captured by MSI within and across animals?

The number of animals was n=3 (10mo) and n=4 (18mo). The number of plaques per animal was N=5-6. We provide this information in the Methods (Animal experimental design) as well as summarize all plaque ROI (MALDI/LCO) and AOI (MALDI/GeoMx) data in a new Supplementary Table 1-5.

b. Figure S1 is insufficiently described to let me understand the structure of the data. A more thorough caption and explanation would be helpful, including a breakdown by source animal rather than just age.

We concur and updated the Supplementary Tables (now Suppl.Tab 1-5) accordingly as detailed in Comment 3a.

c. Could the authors clarify what is shown in Fig. S2b? I am not sure how to interpret these values. It might be more useful to show PCAs instead.

We appreciate this comment. The Figure aims to illustrate the advantage of spatial transcriptomics on the single plaque level vs bulk tissue RNA seq.

Here, the plots aim to illustrate a measure of quantification for synaptic and DAA genes that are significantly changed in response to plaque pathology. The y-axis thereby represents the first principal component of the synaptic and DAA genes, respectively, and serves as a representative abundance value. The results show that spatial analysis allows to delineate plaque specific decrease in synaptic genes and increase in DAA genes, something which is otherwise convoluted in bulk tissue analysis.

We updated the legend (now Supplementary Figure 10) accordingly for improved clarity.

d. Could we have a complete description of all GeoMx FOVs imaged, and their corresponding 15N/14N ratios, morphologies and hyperspectral ratios, anatomical localization, number of unique genes and transcripts sequenced, etc? It would also be important to know the source animals for each FOV.

This could be in the form of a table or heatmap. It will give a better idea of how powered the conclusions are.”

We agree with this suggestion and curate these data in comprehensive Supplementary Fig. 5-6 and Supplementary Tables 3-5.

Minor Comments:

Minor 1: “It might be helpful for the authors to comment further on the applicability of their method in conjunction with higher-resolution spatial transcriptomics methods (e.g., Visium HD, MERFISH, Xenium, etc.) or proteomics (CODEX, IMC, etc.) in future work. Integrating iSILK with the growing landscape of these tools is certainly of interest to many spatial biologists.”

We agree this important point. Unfortunately, the MALDI sample preparation and acquisition impacts tissue morphology and prevents a direct interfacing with other spatial techniques on the same tissue. Similar to the approach used here iSILK can be used on sequential tissues in concert with those techniques.

this is certainly of interest as those other spatial transcriptomics/proteomics methods would give a better resolution of transcripts and proteins towards plaques. As discussed we were faced with the tradeoff between sensitivity and spatial resolution/single cell specificity but agree that follow up studies expanding towards these techniques would be very valuable.

We discuss the compatibility of iSILK with spatial biology techniques in the limitations section.

Minor 2: “Some general commentary on the limitations of their approach might be helpful.”

We included Limitations comment at the end of the Discussion (page 19).

Minor 3: “Fig. 1a scale bar value is not provided. The imaging modality should also be stated.”

We provide this value in the legend.

Minor 4: “As a matter of preference, it might be better for authors to use **perceptually uniform colormaps** instead of ‘spectral’ (e.g., in Fig. 1f). It may be hard for some readers to see.”

We attempted to use a uniform color map instead of spectral coloring, as suggested by the reviewer.

A side-by-side comparison of both color maps (see below) demonstrates that subtle changes in $^{14}\text{N}/^{15}\text{N}$ enrichment are more readily discernible to the human eye employing a spectral color map.

For this reason, we have opted to retain our original color scheme, as it better highlights the features of interest in the data.

Minor 5: “Legends in some panels (e.g., Fig. 1e, 2c) are rather small and hard to see.”

We increased the font size for the respective panels.

Minor 6: “Gene and GO annotations for Fig. 3c–h are also hard to read.”

We increased the font size for the respective panels and rearranged the text to better accommodate the figure dimensions.

Minor 7: “Could the MSI for Fig. 3b be decomposed into 15N and 14N for clarity?”

We provide decomposed images in Fig 3b (as overlay) and in the SI (Supplementary Fig. 3D)

Minor 8: “In ‘spanning across two consecutive 12mm sections’ did the authors mean 12 micron?”

The sections are 12 μm in thickness as rightfully pointed out by the reviewer. This has now been amended accordingly.

Please find attached with this final resubmission our revised manuscript “Isotope Encoded Spatial Biology Identifies Amyloid Plaque-Age-Dependent Structural Maturation and Synaptic Loss” by Woods, Dulewicz et al.

In this revision, we have addressed the remaining comments, including performing minor additional analyses (WCGNA) and making the requested textual and figure modifications. The corresponding changes are detailed in the point-by-point responses below.

With best regards,
Jörg Hanrieder

Reviewer #1:

I have now reviewed the revised manuscript, "Isotope Encoded Spatial Biology Identifies Amyloid Plaque-Age-Dependent Structural Maturation, Synaptic Loss, and Increased Toxicity," along with the authors' point-by-point rebuttal to the concerns I raised in my initial review.

The authors have addressed all of my previous concerns by conducting significant new experiments to validate their findings and strengthen their claims. The new high-sensitivity LC-MS/MS analysis to investigate post-translational modifications and the inclusion of MS/MS spectra for sequence confirmation have fully resolved my initial questions regarding the data.

As a final point of scientific interest, I would like to offer one minor point for the author's consideration. In the MS/MS spectrum for A β 1-42 shown in Supplementary Figure 1F, it is interesting to note the dominance of the b-ion series, while the corresponding y-ion series is almost absent. A brief comment on this observation in the Results section or the caption of the figure could enhance the spectral interpretation of the fragmentation behavior.

Response:

We thank the reviewer for raising this point and added the following statement to the figure description of Supplementary Figure 1: “The b-ion series (charge retained on N-terminal fragment upon peptide fragmentation) dominates in several A β isoforms because the N-terminal region contains multiple charge carriers (e.g., Arg5, Lys16). In contrast, the C-terminal region of A β peptides largely lacks such charge carriers, hence, y-ions are less abundant, and fragment spectra are dominated by the b-ions.”

Reviewer #2:

The authors have made appropriate clarifications of all my previously raised points in

the first round of detailed review and the paper has been improved. Congratulations to very interesting and important work. I support publication as soon as possible.

Response:

We sincerely thank the reviewer for their thorough evaluation of our manuscript and their positive feedback. We are pleased that the revisions addressed the previously raised points.

Reviewer #3:

The presentation of the data in the revised manuscript makes it much more straightforward to read and interpret the data in my opinion. Here are some remaining minor comments for the authors, though they are mostly either presentation issues or nominal analyses that might provide some further insights if the authors choose to perform them. Nevertheless, I believe they have satisfied my major concerns, and I am supportive of publication.

Minor Comments

1. In their discussion of limitations of other high-res ST approaches, they might note that the CosMx 18k now offers near-transcriptome coverage – though at markedly lower gene-wise sensitivity

Response:

We thank the reviewer for this helpful suggestion. We have revised the manuscript to acknowledge that CosMx now provides near whole-transcriptome coverage and have added a note on the associated limitation of reduced gene-wise sensitivity. Please see p20line10-14

2. I would appreciate if the authors could do one additional analysis – to highlight novel gene-gene correlation modules that arise across from analysis across plaques (e.g. with WGCNA or similar).

Response:

We thank the reviewer for this suggestion and performed WGCNA to explore gene-gene correlation networks separately in 10- and 18-month-old mice (see methods section 'Weighted Gene Co-expression Network Analysis (WGCNA)').

For ease of computation and interpretation, we limited the genes included in network construction to those that significantly correlated with our trait of interest, i.e. plaque age.

While the specific filtering criterion was tailored to our study, the use of pre-filtering approaches prior to WGCNA is widely used and described in the literature (Deng et al. 2015, Lin et al. 2018, Zuo et al. 2018).

We have deposited our results (gene cluster membership and GO analysis of each cluster) to the Shiny app, where users can explore genes of interest in a detailed manner.

Please see: https://maciejdulewiczgu.shinyapps.io/MALDI_GEOMX_VOLCANO/

We also included the WGCNA results in a new Supplementary Figure (SI Fig 8), which illustrates the correlation between the identified clusters and plaque age, highlighting selected clusters of interest along with their associated GO terms. The corresponding results are described in the manuscript. In brief, we observed that

gene-gene correlation clusters relating to (glutamatergic) synaptic processes correlated negatively with plaque age, in line with our previously reported results. Additionally, we found clusters enriched in glial proteins (e.g. H2-k1) to be positively correlated with plaque age.

Please see Results: p11line18 and Methods p35

3. The authors might comment on whether their observed gene expression changes signify some T cell infiltration (or any other evidence for this in their animals) versus microglial activation due to overlap in GO annotations. This was what I meant by cryptic expression in my previous review and I apologize for the misunderstanding.

Response:

We thank the reviewer for this clarification and have expanded the Discussion to address the potential contribution of T cell infiltration. We now note that although T cell-associated ontologies were significantly enriched, the driving genes are also broadly expressed in microglia, and canonical T cell markers (CD3, CD4, CD8) were not significantly changed. Therefore, in this dataset, potential T cell infiltration cannot be reliably distinguished from microglial activation. We added a comment in the discussion. p18line11

4. The statement that APP NL-F is strictly better than NL-G-F as it reflects human pathology (Line 387-389) is too strongly made.

Response:

We agree with the reviewer that the original statement was too strong and have revised the text to avoid implying that the APP NL-F model is strictly better or overstating its relevance to human pathology over the NL-G-F model.

5. Scale bars missing in Fig. 3A, 4A.

Response:

We thank the reviewer for noting this omission. Scale bars have now been added to both Fig. 3A and Fig. 4A.

6. Labels for GFAP, Ab, and SYTO colors are missing in Fig. 3B, Supp Fig. 5, 6.

Response:

We thank the reviewer for pointing this out. Labels for GFAP, A β , and SYTO channel colors have now been added to Fig. 3B, Supplementary Fig. 5, and Supplementary Fig. 6.

7. I am not sure why the data must be duplicated as Supp Fig. 3D – would be preferable to instead include these panels in Fig. 3B?

Response:

We agree with the reviewer that duplication was unnecessary. We have re-arranged Fig. 3B to incorporate the relevant panels and have removed Supplementary Fig. 3D to avoid redundancy.

8. The bottom axis labels for Fig. 3C, D are clipped.

Response:

We thank the reviewer for noting this. The bottom axis labels for Fig. 3C and 3D have been corrected to ensure they are fully visible in the revised version of the manuscript.

Typographical

1. Line 190 "oligothiophene (LCO)" > "oligothiophenes (LCOs)
2. Line 429 "nformation" > "information"

Response:

We thank the reviewer for spotting these errors. Both typographical issues have been corrected in the revised manuscript.